# DeepTime: Deep Time-index Meta-learning for Non-stationary Time-series Forecasting

## Abstract

Advances in I.T. infrastructure has led to the collection of longer sequences of time-series. Such sequences are typically non-stationary, exhibiting distribution shifts over time – a challenging scenario for the forecasting task, due to the problems of covariate shift, and conditional distribution shift. In this paper, we show that deep time-index models possess strong synergies with a meta-learning formulation of forecasting, displaying significant advantages over existing neural forecasting methods in tackling the problems arising from non-stationarity. These advantages include having a stronger smoothness prior, avoiding the problem of covariate shift, and having better sample efficiency. To this end, we propose Deep-Time, a deep time-index model trained via meta-learning. Extensive experiments on real-world datasets in the long sequence time-series forecasting setting demonstrate that our approach achieves competitive results with state-of-the-art methods, and is highly efficient. Code is attached as supplementary material, and will be publicly released.

## 1 Introduction

Time-series forecasting has important applications across business and scientific domains, such as demand forecasting (Carbonneau et al., 2008), capacity planning and management (Kim, 2003), and anomaly detection (Laptev et al., 2017). With the advances of I.T. infrastructure, time-series are collected over longer durations, and at a higher sampling frequency. This has led to time-series spanning tens-of-thousands to millions of time steps, on which we would like to perform forecasting on. Such datasets face the unique challenge of non-stationarity, where long sequences face distribution shifts over time, due to factors like concept drift. This has practical implications on forecasting models, which face a degradation in performance at test time (Kim et al., 2021) due to covariate shift, and conditional distribution shift (see Appendix B for formal definitions).

Table 1: Time-index models are defined to be models whose predictions, $\hat{y}_t$, are *purely* functions of the *current* time-index features, $\boldsymbol{\tau}_t$, e.g. relative time-index (1, 2, 3, ...), datetime features (minute-of-hour, week-of-day, etc.). Historical-value models, whose predictions of future time step(s), $\hat{y}_{t+1}$, are explicit functions of past observations, $(\boldsymbol{y}_t, \boldsymbol{y}_{t-1}, \ldots)$, and optionally covariates, $(\boldsymbol{z}_{t+1}, \boldsymbol{z}_t, \boldsymbol{z}_{t-1}, \ldots)$, which can include exogenous time-series or even datetime features.

| Time-index Models | Historical-value Models |
|---|---|
| $\hat{y}_t = f(\boldsymbol{\tau}_t)$ | $\hat{y}_{t+1} = f(\boldsymbol{y}_t, \boldsymbol{y}_{t-1}, \ldots, \boldsymbol{z}_{t+1}, \boldsymbol{z}_t, \ldots)$ |
| **E.g.**: DeepTime, Prophet, Gaussian process | **E.g.**: N-HiTS, Autoformer, DeepAR |

In this work, we posit that deep time-index models exhibit strong synergies with a meta-learning formulation to tackle the problem of non-stationary forecasting, whereas existing neural forecasting methods, which are historical-value models, are unable to take full advantage of this formulation, and are still susceptible to the problem of covariate shift. In the following, we discuss time-index models and their deep counterparts, highlighting how simple deep time-index models are unable to perform forecasting (i.e. extrapolate from historical training data). Yet, endowing them with a meta-learning formulation solves this problem. Thereafter, we demonstrate the advantages of deep time-index meta-learning for non-stationary forecasting and how they alleviate the issues faced by historical-value models, which are namely: (i) meta-learning is an effective solution for conditional distribution shift, (ii) they avoid the problem of covariate shift, (iii) have stronger sample efficiency, and (iv) that time-index models have a stronger smoothness prior.

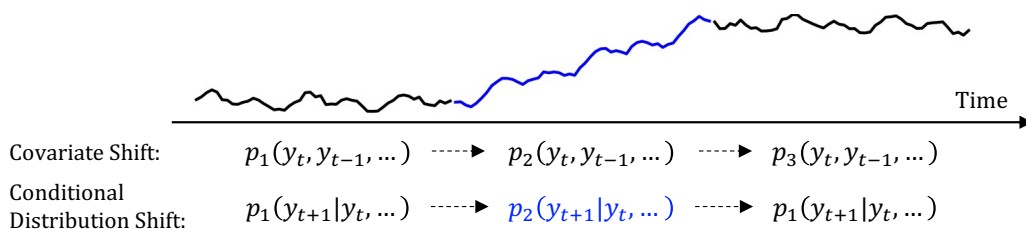

Figure 1: Non-stationary time-series degrade model performance due to covariate shifts and conditional distributional shifts. Such behaviors can be modelled as locally stationary processes, by which contiguous segments are assumed to be stationary. Meta-learning takes advantage of this assumption to adapt to these locally stationary distributions. Yet, existing methods which model the conditional distribution, $p(y_{t+1}|y_t, \ldots)$, are still susceptible to covariate shifts since the meta model takes time-series values as input.

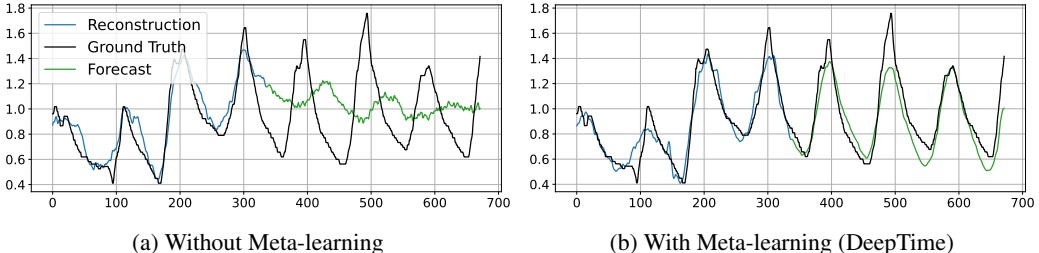

(a) Without Meta-learning          (b) With Meta-learning (DeepTime)

Figure 2: (a) A naive deep time-index model. We visualize a reconstruction of the historical training data, as well as the forecasts. As can be seen, it overfits to historical data and is unable to extrapolate. This model corresponds to (+Local) Table 3 of our ablations. (b) DeepTime, our proposed approach, trained via a meta-learning formulation, successfully learns the appropriate function representation and is able to extrapolate. Visualized here is the last variable of the ETTm2 dataset.

**Deep Time-index Models** On the one hand, classical time-index methods (Taylor & Letham, 2018; Corani et al., 2021; Ord et al., 2017) rely on predefined parametric representation functions $y_t = f(\tau_t) + \varepsilon_t$, where $\varepsilon_t$ represents idiosyncratic changes not accounted for by the model and $f$ could be some polynomial function to represent trend, Fourier series to represent seasonality, or a composition of seasonal-trend components. While these functions are simple and easy to learn, the choice of representation function requires strong domain expertise or computationally heavy cross-validation. Furthermore, predefining the representation function is a strong assumption and may fail under distribution shifts. On the other hand, while deep time-index models (letting $f$ be a deep neural network) present a deceptively clear path to approximate the representation function in a data-driven manner, deep time-index models are too expressive. Trained via straightforward supervised learning on historical values without any inductive bias, they are unable to extrapolate to future time steps (visualized in Figure 2), and a meta-learning formulation is required to do so – this formulation has the added benefit of handling non-stationary forecasting.

**Advantages of Deep Time-index Meta-learning** Firstly, distribution shift of input statistics sharply degrade the prediction accuracy of deep learning models (Nado et al., 2020). Historical-value models, which take past observations as input, suffer from this as an effect of covariate shift. Time-index models easily sidestep this problem since they take time-index features as input. Next, meta-learning is an effective solution to tackle the problem of conditional distribution shift – nearby time steps are assumed to follow a locally stationary distribution (Dahlhaus, 2012; Vogt, 2012) (see Figure 1), considered to be a task. The base learner adapts to this locally stationary distribution, while the meta learner generalizes across various task distributions. In principle, historical-value models are able to take advantage of the meta-learning formulation, however, they still suffer from the problem of covariate shift and sample efficiency issues. Time-index models are also able to achieve greater sample efficiency in the meta-learning formulation. Like many existing state-of-the-art forecasting approaches, time-index models are direct multi-step (DMS) approaches[1]. For a lookback window of length $L$ and forecast horizon of length $H$, a historical-value DMS model requires $N + L + H - 1$ time steps to construct a support set of size $N$, whereas a time-index model only requires $N$ time steps. Not only does this marked increase in sample efficiency mean that

---

[1]DMS methods directly predict the entire forecast horizon, and are contrasted with iterative multi-step (IMS) methods. Further discussion on DMS/IMS, and a taxonomy of forecasting methods in Appendix C.

time-index models can achieve an improved task generalization error bound (Appendix Q), they are also better able to adhere to the assumption of a locally stationary distribution since using more time steps leads to the risk of a non-stationary support set. Finally, time-index models have a stronger smoothness prior (Bengio et al., 2013), i.e. $t \approx t' \implies \boldsymbol{\tau}_t \approx \boldsymbol{\tau}_{t'} \implies f(\boldsymbol{\tau}_t) \approx f(\boldsymbol{\tau}_{t'})$, whereas the complicated parameterization of historical-value models provide no such inductive biases.

To this end, we propose DeepTime, a deep time-index model, endowed with a meta-learning formulation. We leverage implicit neural representations (INR) (Sitzmann et al., 2020b) as our choice of deep time-index models, and also introduce a novel concatenated Fourier features layer to efficiently learn high frequency patterns. The meta-learning formulation is instantiated as a closed-form ridge regressor (Bertinetto et al., 2019) to efficiently tackle the meta-learning formulation. DeepTime manages to overcome the limitations of a naive deep time-index model by learning the appropriate inductive biases for extrapolation over the forecast horizon. It is also more effective than existing historical-value methods at non-stationary time-series forecasting by learning a global meta model shared across tasks and performs adaptation on a locally stationary task, and also sidesteps the covariate shift problem. To summarize, the key contributions of our work are as follows:

- We introduce a novel *forecasting as meta-learning* framework for deep time-index models, enabling them learn the appropriate representation function and tackle the problem of non-stationary forecasting. This is distinct from existing work which leverages meta-learning on historical-value models for adapting to new time-series datasets, where tasks are defined to be the entire time-series (Grazzi et al., 2021).
- We propose DeepTime, leveraging an INR with concatenated Fourier features and closed-form ridge regressor to achieve a highly efficient forecasting model.
- We conduct extensive experiments on the long sequence time-series forecasting (LSTF) datasets, demonstrating DeepTime to be extremely competitive. We perform ablation studies to better understand the contribution of each component of DeepTime, and finally show that it is highly efficient in terms of runtime and memory.

## 2 DEEPTIME

**Problem Formulation** In time-series forecasting, we consider a time-series dataset $(\boldsymbol{y}_1, \boldsymbol{y}_2, \ldots, \boldsymbol{y}_T)$, where $\boldsymbol{y}_t \in \mathbb{R}^m$ is the $m$-dimension observation at time $t$. Given a lookback window $\boldsymbol{Y}_{t-L:t} = [\boldsymbol{y}_{t-L}; \ldots; \boldsymbol{y}_{t-1}]^T \in \mathbb{R}^{L \times m}$ of length $L$, the goal of forecasting is to construct a point forecast over a horizon of length $H$, $\boldsymbol{Y}_{t:t+H} = [\boldsymbol{y}_t; \ldots; \boldsymbol{y}_{t+H-1}]^T \in \mathbb{R}^{H \times m}$. The goal is to learn a time-index model, $f : \mathbb{R} \to \mathbb{R}^m, f : \tau_t \mapsto \hat{\boldsymbol{y}}_t$, where $\tau_t$ is a time-index feature, to quickly adapt to observations in the lookback window, $(\tau_{t-L:t}, \boldsymbol{Y}_{t-L:t})$, by minimizing a reconstruction loss $\mathcal{L} : \mathbb{R}^m \times \mathbb{R}^m \to \mathbb{R}$. Then, we can query it over the forecast horizon to obtain forecasts, $\hat{\boldsymbol{Y}}_{t:t+H} = f(\boldsymbol{\tau}_{t:t+H})$.

In the following, we first describe our *forecasting as meta-learning* framework on time-index models. We emphasize that this formulation falls within the standard time-series forecasting problem and requires no extra information. Next, we further elaborate on our proposed model architecture, and how it uses a differentiable closed-form ridge regression module to efficiently tackle forecasting as meta-learning problem. Psuedocode of DeepTime is available in Appendix E.

### 2.1 FORECASTING AS META-LEARNING

In time-index meta-learning, each lookback window and forecast horizon pair, $(\boldsymbol{Y}_{t-L:t}, \boldsymbol{Y}_{t:t+H})$ is a task. This task yields a single support and query set, which are the lookback window and forecast horizon respectively. Each time coordinate and time-series value pair, $(\tau_{t+i}, \boldsymbol{y}_{t+i})$, is an input-output sample, i.e. $\mathcal{D}^{\mathcal{S}} = \{(\tau_{t-L}, \boldsymbol{y}_{t-L}), \ldots, (\tau_{t-1}, \boldsymbol{y}_{t-1})\}$, $\mathcal{D}^{\mathcal{Q}} = \{(\tau_t, \boldsymbol{y}_t), \ldots, (\tau_{t+H-1}, \boldsymbol{y}_{t+H-1})\}$, where $\tau_{t+i} = \frac{i+L}{L+H-1}$ is a $[0, 1]$-normalized time-index. The time-index model, $f$, is parameterized by $\phi$ and $\theta$, the meta and base parameters respectively, and the bi-level optimization problem can be formalized as:

$$\phi^* = \arg\min_{\phi} \sum_{t=L+1}^{T-H+1} \sum_{j=0}^{H-1} \mathcal{L}(f(\tau_{t+j}; \theta_t^*, \phi), \boldsymbol{y}_{t+j}) \tag{1}$$

$$s.t. \quad \theta_t^* = \arg\min_{\theta} \sum_{j=-L}^{-1} \mathcal{L}(f(\tau_{t+j}; \theta, \phi), \boldsymbol{y}_{t+j}) \tag{2}$$

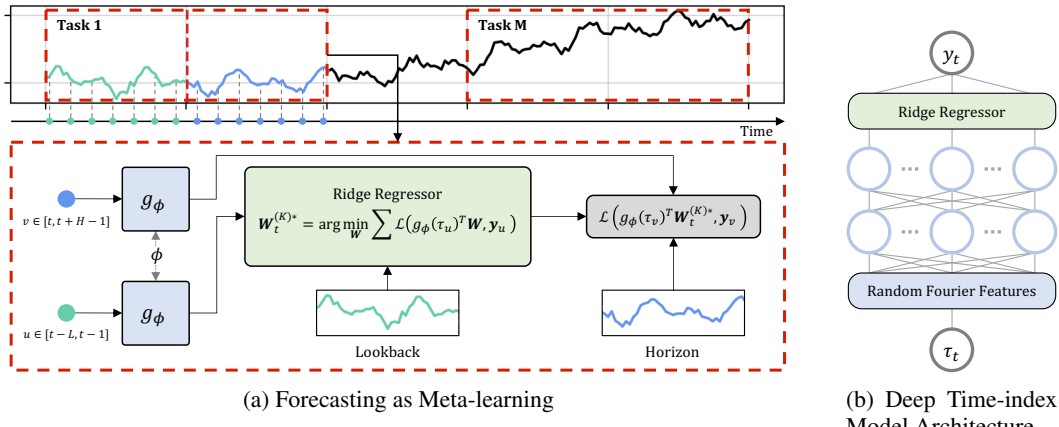

(a) Forecasting as Meta-learning

(b) Deep Time-index Model Architecture

Figure 3: Illustration of DeepTime. A time-series dataset can be split into $M$ tasks as given in the problem formulation. For a given task, the lookback window represents the support set, and the forecast horizon represents the query set. $g_\phi$ represents the meta model associated with the meta parameters. $\phi$ is shared between the lookback window and forecast horizon. Inputs to $g_\phi$ are not normalized due to notation constraints on this illustration. The ridge regressor performs the inner loop optimization, while outer loop optimization is performed over samples from the horizon. As illustrated in Figure 3b, DeepTime has a simple overall architecture, comprising of a random Fourier features layer, an MLP, and a ridge regressor.

Here, the outer summation in Equation (1) over index $t$ represents each lookback-horizon window, corresponding to each task in meta-learning, and the inner summation over index $j$ represents each sample in the query set, or equivalently, each time step in the forecast horizon. The summation in Equation (2) over index $j$ represents each sample in the support set, or each time step in the lookback window. This is illustrated in Figure 3a.

To understand how our meta-learning formulation helps to learn an appropriate function representation from data, we examine how the meta-learning process performs a restriction on hypothesis class of the model $f$. The original hypothesis class of our model, or function representation, $\mathcal{H}_{\text{INR}} = \{f(\tau; \theta, \phi) \mid \theta \in \Theta, \phi \in \Phi\}$, is too large and provides no guarantees that training on the lookback window leads to good extrapolation. The meta-learning formulation allows DeepTime to restrict the hypothesis class of the representation function, from the space of all $K$-layered INRs, to the space of $K$-layered INRs conditioned on the optimal meta parameters, $\mathcal{H}_{\text{DeepTime}} = \{f(\tau; \theta, \phi^*) \mid \theta \in \Theta\}$, where the optimal meta parameters, $\phi^*$, is the minimizer of a forecasting loss (as specified in Equation (1)). Given this hypothesis class, local adaptation is performed over $\mathcal{H}_{\text{DeepTime}}$ given the lookback window, which is assumed to come from a locally stationary distribution, resolving the issue of non-stationarity.

## 2.2 MODEL ARCHITECTURE

**Implicit Neural Representations**  The class of deep models which map coordinates to the value at that coordinate using a stack of multi-layer perceptrons (MLPs) is known as INRs (Sitzmann et al., 2020b; Tancik et al., 2020). We make use a of them as they are a natural fit for time-index models, to map a time-index to the value of the time-series at that time-index. A $K$-layered, ReLU (Nair & Hinton, 2010) INR is a function $f_\theta : \mathbb{R}^c \to \mathbb{R}^m$ which has the following form:

$$\boldsymbol{z}^{(0)} = \boldsymbol{\tau}$$
$$\boldsymbol{z}^{(k+1)} = \max(0, \boldsymbol{W}^{(k)}\boldsymbol{z}^{(k)} + \boldsymbol{b}^{(k)}), \quad k = 0, \ldots, K-1$$
$$f_\theta(\boldsymbol{\tau}) = \boldsymbol{W}^{(K)}\boldsymbol{z}^{(K)} + \boldsymbol{b}^{(K)} \tag{3}$$

where $\boldsymbol{\tau} \in \mathbb{R}^c$ is the time-index. Note that $c = 1$ for our proposed approach as specified in Section 2.1, but we use the notation $\boldsymbol{\tau} \in \mathbb{R}^c$ to allow for generalization to cases where datetime features are included. Tancik et al. (2020) introduced a random Fourier features layer which allows INRs to fit to high frequency functions, by modifying $\boldsymbol{z}^{(0)} = \gamma(\boldsymbol{\tau}) = [\sin(2\pi\boldsymbol{B}\boldsymbol{\tau}), \cos(2\pi\boldsymbol{B}\boldsymbol{\tau})]^T$, where each entry in $\boldsymbol{B} \in \mathbb{R}^{d/2 \times c}$ is sampled from $\mathcal{N}(0, \sigma^2)$ with $d$ is the hidden dimension size of the INR and $\sigma^2$ is the scale hyperparameter. $[\cdot, \cdot]$ is a row-wise stacking operation.

**Concatenated Fourier Features**  While the random Fourier features layer endows INRs with the ability to learn high frequency patterns, one major drawback is the need to perform a hyperparameter sweep for each task and dataset to avoid over or underfitting. We overcome this limitation with a simple scheme of concatenating multiple Fourier basis functions with diverse scale parameters, i.e. $\gamma(\boldsymbol{\tau}) = [\sin(2\pi\boldsymbol{B}_1\boldsymbol{\tau}), \cos(2\pi\boldsymbol{B}_1\boldsymbol{\tau}), \ldots, \sin(2\pi\boldsymbol{B}_S\boldsymbol{\tau}), \cos(2\pi\boldsymbol{B}_S\boldsymbol{\tau})]^T$, where elements in $\boldsymbol{B}_s \in \mathbb{R}^{d/2 \times c}$ are sampled from $\mathcal{N}(0, \sigma_s^2)$, and $\boldsymbol{W}^{(0)} \in \mathbb{R}^{d \times Sd}$. We perform an analysis in Section 3.3 and show that the performance of our proposed Concatenated Fourier Features (CFF) does not significantly deviate from the setting with the optimal scale parameter obtained from a hyperparameter sweep.

**Differentiable Closed-form Solvers**  One key aspect to tackling forecasting as a meta-learning problem is efficiency. Optimization-based meta-learning approaches originally perform an expensive bi-level optimization procedure on the entire neural network model by backpropagating through inner gradient steps (Ravi & Larochelle, 2017; Finn et al., 2017). Since each forecast is now treated as an inner loop optimization problem, it needs to be sufficiently fast to be competitive with competing methods. We achieve this by restricting the inner loop optimization to only apply to the last layer of the INR. As a result, we can perform the inner loop optimization on this linear layer using the closed-form solution of a ridge regressor for the case of mean squared error loss. We note that our formulation is general, and any differentiable solver can be used instead (Bertinetto et al., 2019). This means that for a $K$-layered model, $\phi = \{\boldsymbol{W}^{(0)}, \boldsymbol{b}^{(0)}, \ldots, \boldsymbol{W}^{(K-1)}, \boldsymbol{b}^{(K-1)}, \lambda\}$ are the meta parameters and $\theta = \{\boldsymbol{W}^{(K)}\}$ are the base parameters, following notation from Equation (3). Then let $g_\phi : \mathbb{R} \to \mathbb{R}^d$ be the meta learner where $g_\phi(\boldsymbol{\tau}) = \boldsymbol{z}^{(K)}$. For task $t$ with the corresponding lookback-horizon pair, $(\boldsymbol{Y}_{t-L:t}, \boldsymbol{Y}_{t:t+H})$, the support set features obtained from the meta learner is denoted $\boldsymbol{Z}_{t-L:t} = [g_\phi(\boldsymbol{\tau}_{t-L}); \ldots; g_\phi(\boldsymbol{\tau}_{t-1})]^T \in \mathbb{R}^{L \times d}$, where $[\cdot; \cdot]$ is a column-wise concatenation operation. The inner loop thus solves the optimization problem:

$$\boldsymbol{W}_t^{(K)*} = \underset{\boldsymbol{W}}{\arg\min} \, ||\boldsymbol{Z}_{t-L:t}\boldsymbol{W} - \boldsymbol{Y}_{t-L:t}||^2 + \lambda||\boldsymbol{W}||^2$$
$$= (\boldsymbol{Z}_{t-L:t}^T \boldsymbol{Z}_{t-L:t} + \lambda\boldsymbol{I})^{-1} \boldsymbol{Z}_{t-L:t}^T \boldsymbol{Y}_{t-L:t} \tag{4}$$

Now, let $\boldsymbol{Z}_{t:t+H} = [g_\phi(\boldsymbol{\tau}_t); \ldots; g_\phi(\boldsymbol{\tau}_{t+H-1})]^T \in \mathbb{R}^{H \times d}$ be the query set features. Then, our predictions are $\hat{\boldsymbol{Y}}_{t:t+H} = \boldsymbol{Z}_{t:t+H}\boldsymbol{W}_t^{(K)*}$. This closed-form solution is differentiable, which enables gradient updates on the parameters of the meta learner, $\phi$. A bias term can be included for the closed-form ridge regressor by appending a scalar 1 to the feature vector $g_\phi(\boldsymbol{\tau})$. The end result of training DeepTime on a dataset is the restricted hypothesis class $\mathcal{H}_{\text{DeepTime}} = \left\{ g_{\phi^*}(\boldsymbol{\tau})^T \boldsymbol{W}^{(K)} \mid \boldsymbol{W}^{(K)} \in \mathbb{R}^{d \times m} \right\}$. This is illustrated in Figure 3b.

Some confusion regarding DeepTime's categorization as a time-index model may arise from the above simplified equation for predictions, since forecasts are now a function the lookback window due to the closed-form solution of $\boldsymbol{W}_t^{(K)*}$. However, we highlight that DeepTime is a meta-learning algorithm on top of a deep time-index model – it comprises a *learning algorithm*, $\mathcal{A} : \mathcal{H} \times \mathbb{R}^{L \times m} \to \mathcal{H}$, specified in Equation (2) (the inner loop optimization step), and the deep time-index model itself, $f \in \mathcal{H}_{\text{DeepTime}}$. Thus, forecasts are of the form, $\hat{\boldsymbol{y}}_{t+h} = \mathcal{A}(f, \boldsymbol{Y}_{t-L:t})(\boldsymbol{\tau}_{t+h})$, and as can be seen, while the inner loop optimization step is a function of past observations, the adapted time-index model it yields is purely a function of time-index features. Further discussion can be found in Appendix D.

## 3 EXPERIMENTS

We evaluate DeepTime on both synthetic datasets, and a variety of real-world data. We ask the following questions: (i) Is DeepTime, trained on a family of functions following the same parametric form, able to perform extrapolation on unseen functions? (ii) How does DeepTime compare to other forecasting models on real-world data? (iii) What are the key contributing factors to the good performance of DeepTime?

### 3.1 EXPERIMENTS ON SYNTHETIC DATA

We first consider DeepTime's ability to extrapolate on the following functions specified by some parametric form: (i) the family of linear functions, $y = ax + b$, (ii) the family of cubic functions, $y = ax^3 + bx^2 + cx + d$, and (iii) sums of sinusoids, $\sum_j A_j \sin(\omega_j x + \varphi_j)$. Parameters of the functions

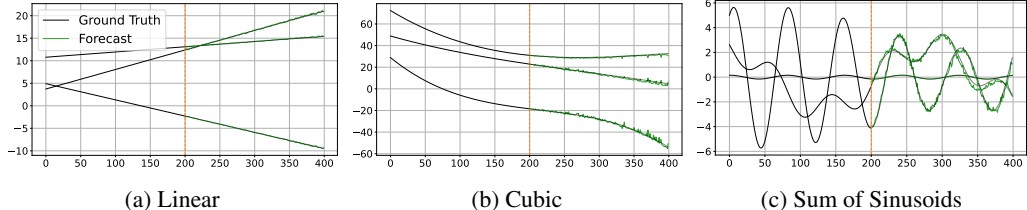

|  | (a) Linear | (b) Cubic | (c) Sum of Sinusoids |

Figure 4: Predictions of DeepTime on three **unseen** functions for each function class. The orange line represents the split between lookback window and forecast horizon.

Table 2: Multivariate forecasting benchmark on long sequence time-series forecasting. Best results are highlighted in **bold**, and second best results are underlined.

| Methods | | DeepTime | | NS Transformer | | N-HiTS | | ETSformer | | FEDformer | | Autoformer | | Informer | | LogTrans | | GP | |
|---|---|---|---|---|---|---|---|---|---|---|---|---|---|---|---|---|---|---|---|
| Metrics | | MSE | MAE | MSE | MAE | MSE | MAE | MSE | MAE | MSE | MAE | MSE | MAE | MSE | MAE | MSE | MAE | MSE | MAE |
| ETTm2 | 96 | **0.166** | 0.257 | 0.192 | 0.274 | 0.176 | 0.255 | 0.189 | 0.280 | 0.203 | 0.287 | 0.255 | 0.339 | 0.365 | 0.453 | 0.768 | 0.642 | 0.442 | 0.422 |
| | 192 | **0.225** | **0.302** | 0.280 | 0.339 | 0.245 | 0.305 | 0.253 | 0.319 | 0.269 | 0.328 | 0.281 | 0.340 | 0.533 | 0.563 | 0.989 | 0.757 | 0.605 | 0.505 |
| | 336 | **0.277** | **0.336** | 0.334 | 0.361 | 0.295 | 0.346 | 0.314 | 0.357 | 0.325 | 0.366 | 0.339 | 0.372 | 1.363 | 0.887 | 1.334 | 0.872 | 0.731 | 0.569 |
| | 720 | **0.383** | **0.409** | 0.417 | 0.413 | 0.401 | 0.426 | 0.414 | 0.413 | 0.421 | 0.415 | 0.422 | 0.419 | 3.379 | 1.388 | 3.048 | 1.328 | 0.959 | 0.669 |
| ECL | 96 | **0.137** | **0.238** | 0.169 | 0.273 | 0.147 | 0.249 | 0.187 | 0.304 | 0.183 | 0.297 | 0.201 | 0.317 | 0.274 | 0.368 | 0.258 | 0.357 | 0.503 | 0.538 |
| | 192 | **0.152** | **0.252** | 0.182 | 0.286 | 0.167 | 0.269 | 0.199 | 0.315 | 0.195 | 0.308 | 0.222 | 0.334 | 0.296 | 0.386 | 0.266 | 0.368 | 0.505 | 0.543 |
| | 336 | **0.166** | **0.268** | 0.200 | 0.304 | 0.186 | 0.290 | 0.212 | 0.329 | 0.212 | 0.313 | 0.231 | 0.338 | 0.300 | 0.394 | 0.280 | 0.380 | 0.612 | 0.614 |
| | 720 | **0.201** | **0.302** | 0.222 | 0.321 | 0.243 | 0.340 | 0.233 | 0.345 | 0.231 | 0.343 | 0.254 | 0.361 | 0.373 | 0.439 | 0.283 | 0.376 | 0.652 | 0.635 |
| Exchange | 96 | **0.081** | 0.205 | 0.111 | 0.237 | 0.092 | 0.211 | 0.085 | **0.204** | 0.139 | 0.276 | 0.197 | 0.323 | 0.847 | 0.752 | 0.968 | 0.812 | 0.136 | 0.267 |
| | 192 | **0.151** | **0.284** | 0.219 | 0.335 | 0.208 | 0.322 | 0.182 | 0.303 | 0.256 | 0.369 | 0.300 | 0.369 | 1.204 | 0.895 | 1.040 | 0.851 | 0.229 | 0.348 |
| | 336 | **0.314** | **0.412** | 0.421 | 0.476 | 0.371 | 0.443 | 0.348 | 0.428 | 0.426 | 0.464 | 0.509 | 0.524 | 1.672 | 1.036 | 1.659 | 1.081 | 0.372 | 0.447 |
| | 720 | **0.856** | **0.663** | 1.092 | 0.769 | 0.888 | 0.723 | 1.025 | 0.774 | 1.090 | 0.800 | 1.447 | 0.941 | 2.478 | 1.310 | 1.941 | 1.127 | 1.135 | 0.810 |
| Traffic | 96 | **0.390** | **0.275** | 0.612 | 0.338 | 0.402 | 0.282 | 0.607 | 0.392 | 0.562 | 0.349 | 0.613 | 0.388 | 0.719 | 0.391 | 0.684 | 0.384 | 1.112 | 0.665 |
| | 192 | **0.402** | **0.278** | 0.613 | 0.340 | 0.420 | 0.297 | 0.621 | 0.399 | 0.562 | 0.346 | 0.616 | 0.382 | 0.696 | 0.379 | 0.685 | 0.390 | 1.133 | 0.671 |
| | 336 | **0.415** | **0.288** | 0.618 | 0.328 | 0.448 | 0.313 | 0.622 | 0.396 | 0.570 | 0.323 | 0.622 | 0.337 | 0.777 | 0.420 | 0.733 | 0.408 | 1.274 | 0.723 |
| | 720 | **0.449** | **0.307** | 0.653 | 0.355 | 0.539 | 0.353 | 0.632 | 0.396 | 0.596 | 0.368 | 0.660 | 0.408 | 0.864 | 0.472 | 0.717 | 0.396 | 1.280 | 0.719 |
| Weather | 96 | 0.166 | 0.221 | 0.173 | 0.223 | 0.158 | 0.195 | 0.197 | 0.281 | 0.217 | 0.296 | 0.266 | 0.336 | 0.300 | 0.384 | 0.458 | 0.490 | 0.395 | 0.356 |
| | 192 | **0.207** | 0.261 | 0.245 | 0.285 | 0.211 | 0.247 | 0.237 | 0.312 | 0.276 | 0.336 | 0.307 | 0.367 | 0.598 | 0.544 | 0.658 | 0.589 | 0.450 | 0.398 |
| | 336 | **0.251** | **0.298** | 0.321 | 0.338 | 0.274 | 0.300 | 0.298 | 0.353 | 0.339 | 0.380 | 0.359 | 0.359 | 0.578 | 0.523 | 0.797 | 0.652 | 0.508 | 0.440 |
| | 720 | **0.301** | **0.338** | 0.414 | 0.410 | 0.351 | 0.353 | 0.352 | 0.388 | 0.403 | 0.428 | 0.419 | 0.419 | 1.059 | 0.741 | 0.869 | 0.675 | 0.498 | 0.450 |
| ILI | 24 | 2.425 | 1.086 | 2.294 | 0.945 | **1.862** | **0.869** | 2.527 | 1.020 | 2.203 | 0.963 | 3.483 | 1.287 | 5.764 | 1.677 | 4.480 | 1.444 | 2.331 | 1.036 |
| | 36 | 2.231 | 1.008 | **1.825** | **0.848** | 2.071 | 0.969 | 2.615 | 1.007 | 2.272 | 0.976 | 3.103 | 1.148 | 4.755 | 1.467 | 4.799 | 1.467 | 2.167 | 1.002 |
| | 48 | 2.230 | 1.016 | **2.010** | **0.900** | 2.346 | 1.042 | 2.359 | 0.972 | 2.209 | 0.981 | 2.669 | 1.085 | 4.763 | 1.469 | 4.800 | 1.468 | 2.961 | 1.180 |
| | 60 | **2.143** | 0.985 | 2.178 | **0.963** | 2.560 | 1.073 | 2.487 | 1.016 | 2.545 | 1.061 | 2.770 | 1.125 | 5.264 | 1.564 | 5.278 | 1.560 | 3.108 | 1.214 |

(i.e. $a, b, c, d, A_j, \omega_j, \varphi_j$) are sampled randomly (further details in Appendix F) to construct distinct tasks. A total of 400 time steps are sampled, with a lookback window length of 200 and forecast horizon of 200. Figure 4 demonstrates that DeepTime is able to perform extrapolation on unseen test functions/tasks after being trained via our meta-learning formulation. It demonstrates an ability to approximate and adapt, based on the lookback window, linear and cubic polynomials, and even sums of sinusoids. Next, we evaluate DeepTime on real-world datasets, against state-of-the-art baselines.

## 3.2 EXPERIMENTS ON REAL-WORLD DATA

Experiments are performed on 6 real-world datasets – Electricity Transformer Temperature (ETT), Electricity Consuming Load (ECL), Exchange, Traffic, Weather, and Influenza-like Illness (ILI) with full details in Appendix G. We evaluate the performance of our proposed approach using two metrics, the mean squared error (MSE) and mean absolute error (MAE) metrics. The datasets are split into train, validation, and test sets chronologically, following a 70/10/20 split for all datasets except for *ETTm2* which follows a 60/20/20 split, as per convention. The univariate benchmark selects the last index of the multivariate dataset as the target variable, following previous work (Xu et al., 2021). Preprocessing on the data is performed by standardization based on train set statistics. Hyperparameter selection is performed on only one value, the lookback length multiplier, $L = \mu*H$, which decides the length of the lookback window. We search through the values $\mu = [1, 3, 5, 7, 9]$, and select the best value based on the validation loss. Further implementation details on DeepTime are reported in Appendix H, and detailed hyperparameters are reported in Appendix I. Reported results for DeepTime are averaged over three runs, and standard deviation is reported in Appendix K.

**Results**   We compare DeepTime to the following baselines for the multivariate setting, N-HiTS (Challu et al., 2022), ETSformer (Woo et al., 2022), Fedformer (Zhou et al., 2022) (we report the best score for each setting from the two variants they present), Autoformer (Xu et al., 2021), Informer (Zhou et al., 2021), LogTrans (Li et al., 2019), Non-stationary (NS) Transformer (Liu et al.,

Table 3: Ablation study on variants of DeepTime. Starting from the original version, we add (+) or remove (-) some component from DeepTime. *RR* stands for the differentiable closed-form **r**idge **r**egressor, removing it refers to replacing this module with a simple linear layer trained via gradient descent across all training samples (i.e. without meta-learning formulation). *Local* refers to training an INR from scratch via gradient descent for each lookback window (RR is **not** used here, and there is no training phase). *Datetime* refers to datetime features. Further model details can be found in Appendix O.1.

| Methods | DeepTime | | + Datetime | | - RR | | - RR + Datetime | | + Local | | + Local + Datetime | |
|---|---|---|---|---|---|---|---|---|---|---|---|---|
| Metrics | MSE | MAE | MSE | MAE | MSE | MAE | MSE | MAE | MSE | MAE | MSE | MAE |
| ETTm2 96 | **0.166** | **0.257** | 0.226 | 0.303 | 3.072 | 1.345 | 3.393 | 1.400 | 0.251 | 0.331 | 0.250 | 0.327 |
| 192 | **0.225** | **0.302** | 0.309 | 0.362 | 3.064 | 1.343 | 3.269 | 1.381 | 0.322 | 0.371 | 0.323 | 0.366 |
| 336 | **0.277** | **0.336** | 0.341 | 0.381 | 2.920 | 1.309 | 3.442 | 1.401 | 0.370 | 0.412 | 0.367 | 0.396 |
| 720 | **0.383** | **0.409** | 0.453 | 0.447 | 2.773 | 1.273 | 3.400 | 1.399 | 0.443 | 0.449 | 0.455 | 0.461 |

Table 4: Ablation study on backbone models. DeepTime refers to our proposed approach, an INR with random Fourier features sampled from a range of scales. MLP refers to replacing the random Fourier features with a linear map from input dimension to hidden dimension. SIREN refers to an INR with periodic activations as proposed by Sitzmann et al. (2020b). RNN refers to an autoregressive recurrent neural network (inputs are the time-series values, $y_t$). All approaches include differentiable closed-form ridge regressor. Further model details can be found in Appendix O.2.

| Methods | DeepTime | | MLP | | SIREN | | RNN | |
|---|---|---|---|---|---|---|---|---|
| Metrics | MSE | MAE | MSE | MAE | MSE | MAE | MSE | MAE |
| ETTm2 96 | **0.166** | **0.257** | 0.186 | 0.284 | 0.236 | 0.325 | 0.233 | 0.324 |
| 192 | **0.225** | **0.302** | 0.265 | 0.338 | 0.295 | 0.361 | 0.275 | 0.337 |
| 336 | **0.277** | **0.336** | 0.316 | 0.372 | 0.327 | 0.386 | 0.344 | 0.383 |
| 720 | **0.383** | **0.409** | 0.401 | 0.417 | 0.438 | 0.453 | 0.431 | 0.432 |

2022), and Gaussian Process (GP) (Rasmussen, 2003). For the univariate setting, we include additional univariate forecasting models, N-BEATS (Oreshkin et al., 2020), DeepAR (Salinas et al., 2020), Prophet (Taylor & Letham, 2018), and ARIMA. Baseline results are obtained from the respective papers. Table 2 and Table 9 (in Appendix J for space) summarizes the multivariate and univariate forecasting results respectively. DeepTime achieves state-of-the-art performance on 20 out of 24 settings in MSE, and 17 out of 24 settings in MAE on the multivariate benchmark, and also achieves competitive results on the univariate benchmark despite its simple architecture compared to the baselines comprising complex fully connected architectures and computationally intensive Transformer architectures.

## 3.3 ABLATION STUDIES

We perform an ablation study to understand how various training schemes and input features affect the performance of DeepTime. Table 3 presents these results. First, we observe that our meta-learning formulation is a critical component to the success of DeepTime. We note that DeepTime without meta-learning may not be a meaningful baseline since the model outputs are always the same regardless of the input lookback window. Including datetime features helps alleviate this issue, yet we observe that the inclusion of datetime features generally lead to a degradation in performance. In the case of DeepTime, we observed that the inclusion of datetime features lead to a much lower training loss, but degradation in test performance – this is a case of meta-learning memorization (Yin et al., 2020) due to the tasks becoming non-mutually exclusive (Rajendran et al., 2020). Finally, we observe that the meta-learning formulation is indeed superior to training a model from scratch for each lookback window.

In Table 4 we perform an ablation study on various backbone architectures, while retaining the differentiable closed-form ridge regressor. We observe a degradation when the random Fourier features layer is removed, due to the spectral bias problem which neural networks face (Rahaman et al., 2019; Tancik et al., 2020). DeepTime outperforms the SIREN variant of INRs which is consistent with observations INR literature. Finally DeepTime outperforms the RNN variant which is the model proposed in Grazzi et al. (2021). This is a direct comparison between IMS historical-value models and time-index models, and highlights the benefits of a time-index models.

Table 5: Comparison of CFF against the optimal and pessimal scales as obtained from the hyperparameter sweep. We also calculate the change in performance between CFF and the optimal and pessimal scales, where a positive percentage refers to a CFF underperforming, and negative percentage refers to CFF outperforming, calculated as % change $= (\text{MSE}_{CFF} - \text{MSE}_{Scale})/\text{MSE}_{Scale}$.

| | | CFF | | Optimal Scale (% change) | | Pessimal Scale (% change) | |
|---|---|---|---|---|---|---|---|
| Metrics | | MSE | MAE | MSE | MAE | MSE | MAE |
| ETTm2 | 96 | 0.166 | 0.257 | 0.164 (1.20%) | 0.257 (-0.05%) | 0.216 (-23.22%) | 0.300 (-14.22%) |
| | 192 | 0.225 | 0.302 | 0.220 (1.87%) | 0.301 (0.25%) | 0.275 (-18.36%) | 0.340 (-11.25%) |
| | 336 | 0.277 | 0.336 | 0.275 (0.70%) | 0.336 (-0.22%) | 0.340 (-18.68%) | 0.375 (-10.57%) |
| | 720 | 0.383 | 0.409 | 0.364 (5.29%) | 0.392 (4.48%) | 0.424 (-9.67%) | 0.430 (-4.95%) |

(a) Runtime Analysis

(b) Memory Analysis

Figure 5: Computational efficiency benchmark on the ETTm2 multivariate dataset, on a batch size of 32. Runtime is measured for one iteration (forward + backward pass). Left: Runtime/Memory usage as lookback length varies, horizon is fixed to 48. Right: Runtime/Memory usage as horizon length varies, lookback length is fixed to 48. Further model details can be found in Appendix P.

Lastly, we perform a comparison between the optimal and pessimal scale hyperparameter for the vanilla random Fourier features layer, against our proposed CFF. We first report the results on each scale hyperparameter for the vanilla random Fourier features layer in Table 13, Appendix N. As with the other ablation studies, the results reported in Table 13 is based on performing a hyperparameter sweep across lookback length multiplier, and selecting the optimal settings based on the validation set, and reporting the test set results. Then, the optimal and pessimal scales are simply the best and worst results based on Table 13. Table 5 shows that CFF achieves extremely low deviation from the optimal scale across all settings, yet retrains the upside of avoiding this expensive hyperparameter tuning phase. We also observe that tuning the scale hyperparameter is extremely important, as CFF obtains up to a 23.22% improvement in MSE over the pessimal scale hyperparameter.

## 3.4 COMPUTATIONAL EFFICIENCY

Finally, we analyse DeepTime's efficiency in both runtime and memory usage, with respect to both lookback window and forecast horizon lengths. The main bottleneck in computation for DeepTime is the matrix inversion operation in the ridge regressor, canonically of $\mathcal{O}(n^3)$ complexity. This is a major concern for DeepTime as $n$ is linked to the length of the lookback window. As mentioned in Bertinetto et al. (2019), the Woodbury formulation,

$$\boldsymbol{W}^* = \boldsymbol{Z}^T(\boldsymbol{Z}\boldsymbol{Z}^T + \lambda\boldsymbol{I})^{-1}\boldsymbol{Y}$$

is used to alleviate the problem, leading to an $\mathcal{O}(d^3)$ complexity, where $d$ is the hidden size hyperparameter, fixed to some value (see Appendix I). Figure 5 demonstrates that DeepTime is highly efficient, even when compared to efficient Transformer models, recently proposed for the long sequence time-series forecasting task, as well as fully connected models.

## 4 RELATED WORK

**Neural Forecasting**  Neural forecasting Benidis et al. (2020) methods have seen great success in recent times. One related line of research are Transformer-based methods for LSTF (Li et al., 2019; Zhou et al., 2021; Xu et al., 2021; Woo et al., 2022; Zhou et al., 2022) which aim to not only achieve high accuracy, but to overcome the vanilla attention's quadratic complexity. Fully connected methods (Oreshkin et al., 2020; Challu et al., 2022) have also shown success, with Challu et al. (2022) introducing hierarchical interpolation and multi-rate data sampling for the LSTF task. Meta-learning with the use of a differentiable closed-form solver has been explored in time-series forecasting (Grazzi et al., 2021), but for the meta-forecasting setting which adapts to new time-series datasets rather than to tackle non-stationarity, using an IMS historical-value backbone model.

**Time-index Models**  Time-index models take as input time-index features such as datetime features to predict the value of the time-series at that time step. They have been well explored as a special case of regression analysis (Hyndman & Athanasopoulos, 2018; Ord et al., 2017), and many different predictors have been proposed for the classical setting,including linear, polynomial, and piecewise linear trends, and dummy variables indicating holidays. Of note, Fourier terms have been used to model periodicity, or seasonal patterns, and is also known as harmonic regression (Young et al., 1999). Prophet (Taylor & Letham, 2018) is a popular classical approach which uses a structural time-series formulation, specialized for business forecasting. Another classical approach of note are Gaussian Processes (Rasmussen, 2003; Corani et al., 2021) which are non-parametric models, often requiring complex kernel engineering. Godfrey & Gashler (2017) introduced an initial attempt at using time-index based neural networks to fit a time-series for forecasting. Yet, their work is more reminiscent of classical methods, manually specifying periodic and non-periodic activation functions, analogous to the representation functions.

**Implicit Neural Representations**  INRs have recently gained popularity in the area of neural rendering (Tewari et al., 2021). They parameterize a signal as a continuous function, mapping a coordinate to the value at that coordinate. A key finding was that positional encodings (Mildenhall et al., 2020; Tancik et al., 2020) are critical for ReLU MLPs to learn high frequency details, while another line of work introduced periodic activations (Sitzmann et al., 2020b). Meta-learning on via INRs have been explored for various data modalities, typically over images or for neural rendering tasks (Sitzmann et al., 2020a; Tancik et al., 2021; Dupont et al., 2021), using both hypernetworks and optimization-based approaches. Yüce et al. (2021) show that meta-learning on INRs is analogous to dictionary learning. In time-series, Jeong & Shin (2022) explored using INRs for anomaly detection, opting to make use of periodic activations and temporal positional encodings.

## 5 DISCUSSION

In this paper, we proposed DeepTime, a deep time-index based model trained via a meta-learning formulation to automatically learn a representation function from time-series data, rather than manually defining the representation function as per classical methods. The meta-learning formulation further enables DeepTime to be utilized for non-stationary time-series by adapting to the locally stationary distribution. Importantly, we use a closed-form ridge regressor to tackle the meta-learning formulation to ensure that predictions are computationally efficient. Our extensive empirical analysis shows that DeepTime, while being a much simpler model architecture compared to prevailing state-of-the-art methods, achieves competitive performance across forecasting benchmarks on real world datasets. We perform substantial ablation studies to identify the key components contributing to the success of DeepTime, and also show that it is highly efficient.

**Limitations & Future Work**  Despite having verified DeepTime's effectiveness, we expect some under-performance in cases where the lookback window contains significant anomalies, or an abrupt change point which violates the locally stationary assumption. Next, while out of scope for our current work, a limitation that DeepTime faces is that it does not consider holidays and events. We leave the consideration of such features as a potential future direction, along with the incorporation of exogenous covariates and datetime features, whilst avoiding the incursion of the meta-learning memorization problem. Finally, time-index models are a natural fit for missing value imputation, as well as other time-series intelligence tasks for irregular time-series – this is another interesting future direction to extend deep time-index models towards.

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

## A    EXTENDED RELATED WORK

**Non-stationarity and Distribution Shift**    Switching state-space models (Ghahramani & Hinton, 2000) generalizes and combines hidden Markov models and state-space models, where the dynamics in each regime is typically represented by a linear model (linear dynamic system) and switches between regimes, controlled by hidden transition probabilities of a Markov process. They can be applied to non-stationary time-series, but makes additional assumptions that the time-series has a predefined number of regimes or generating distributions. Sequential Neural Processes (Singh et al., 2019) incorporates a temporal state-transition model of stochastic processes, extending Neural Process framework to dynamic stochastic processes. Importantly, SNPs use a "black-box meta-learning" approach, while DeepTime uses an "optimization-based meta-learning" approach. Further differences include that the standard SNP setting requires knowledge of task boundaries, and multiple support/query sets per task. Du et al. (2021) tackled temporal covariate shift by distribution matching, an approach popularly used in domain adaptation. They introduce the Temporal Distribution Characterization module which divides a given time-series into regions with different disribution, and a Temporal Distribution Matching module reduces distribution mismatch in the time-series. Their approach is built on top of an RNN architecture. Kim et al. (2021) introduced a learnable instance normalization method to tackle the covariate shift problem. Their approach is ad-hoc and can be attached to any existing architecture. Most relevant to our work is Non-stationary Transformers (Liu et al., 2022), which introduced an instance normalization method for the LSTF task. Rather than proposing a generic module, their approach is specialized for Transformer-based architectures and performs normalization on each layer to tackle the non-stationarity of intermediate representations, rather than just the inputs and outputs.

## B    NON-STATIONARITY AND DISTRIBUTION SHIFT

In our work, we tackle the problem of non-stationarity in time-series data, which has been a well explored problem in the context of classical time-series analysis. We map this problem to the modern setting of deep learning for time-series forecasting. As mentioned in Section 1, long time-series datasets collected due to the advances of I.T. infrastructure has been plagued by the problem of non-stationarity. In particular, we tackle the problems of *covariate shift*, and *conditional distribution shift* which arise from it.

**Definition 1.** *(Covariate Shift) Given a stochastic process $\{Y_t\}_{t=1}^T$ and let $p(y_t, y_{t-1}, \ldots, y_{t-L+1})$ be the unconditional joint distribution of a length $L$ segment. The stochastic process is said to experience covariate shift if any two segments are drawn from different distributions, i.e. $p(y_t, y_{t-1}, \ldots, y_{t-L}) \neq p(y_{t'}, y_{t'-1}, \ldots, y_{t'-L}), \forall\, t \neq t'.$*

**Definition 2.** *(Conditional Distribution Shift) Given a stochastic process $\{Y_t\}_{t=1}^T$ and let $p(y_{t+1}|y_t, y_{t-1}, \ldots, y_{t-L+1})$ be the conditional distribution of $Y_{t+1}$ on a length $L$ segment of previous time steps. The stochastic process is said to experience a conditional distribution shift if any two segments have different conditional distributions, i.e. $p(y_{t+1}|y_t, y_{t-1}, \ldots, y_{t-L+1}) \neq p(y_{t'+1}|y_{t'}, y_{t'-1}, \ldots, y_{t'-L+1}), \forall\, t \neq t'$*

## C   CATEGORIZATION OF FORECASTING METHODS

Table 6: Categorization of time-series forecasting methods over the dimensions of time-index vs historical-value methods, and DMS vs IMS methods.

|  | **Time-index** | **Historical-value** |
|---|---|---|
| **DMS** | DeepTime
Prophet
Gaussian process
Time-series regression | N-HiTS
FEDformer
ETSformer
Autoformer
Informer
N-BEATS |
| **IMS** | - | DeepAR
ARIMA
ETS |

**Multi-step Forecasts**   Forecasting over a horizon (multiple time steps) can be achieved via two strategies, direct multi-step, or iterative multi-step (Marcellino et al., 2006; Chevillon, 2007; Taieb et al., 2012), or even a mixture of both, but this has been less explored:

- **Direct Multi-step (DMS)**: A DMS forecaster directly predicts forecasts for the entire horizon. For example, to achieve a multi-step forecast of $H$ time steps, a DMS forecaster simply outputs $H$ values in a single forward pass.

- **Iterative Multi-step (IMS)**: An IMS forecaster iteratively predicts one step ahead, and consumes this forecast to make a subsequent prediction. This is performed iteratively, until the desired length is achieved.

## D   FURTHER DISCUSSION ON DEEPTIME AS A TIME-INDEX MODEL

We first reiterate our definitions of time-index and historical-value models from Section 1. Time-index models are models whose predictions are *purely* functions of *current* time-index features. To perform forecasting (i.e. make predictions over some forecast horizon), time-index models make the predictions $\hat{\boldsymbol{y}}_{t+h} = f(\boldsymbol{\tau}_{t+h})$ for $h = 0, \ldots, H-1$. Historical-value models predict the time-series value of future time step(s) as a function of past observations, and optionally, covariates.

| **Time-index Models** | **Historical-value Models** |
|---|---|
| $\hat{\boldsymbol{y}}_t = f(\boldsymbol{\tau}_t)$ | $\hat{\boldsymbol{y}}_{t+1} = f(\boldsymbol{y}_t, \boldsymbol{y}_{t-1}, \ldots, \boldsymbol{z}_{t+1}, \boldsymbol{z}_t, \ldots)$ |

Next, we further discuss some subtleties of how time-index models interact with past observations. Astute readers may have noticed DeepTime to be a function of the past observations. In particular, that Equations (3) and (4) indicate that forecasts from DeepTime are in fact linear in the lookback window. However, we highlight that this is not in contradiction with our definition of time-index and historical-value models. Here, we differentiate between the *model*, $f$ and the *learning algorithm*, $\mathcal{A}$. The learning algorithm $\mathcal{A} : \mathcal{H} \times \mathbb{R}^{L \times m} \to \mathcal{H}$ takes as input a model from the hypothesis class $\mathcal{H}$ and, past observations, returning a model minimizing the loss function $\mathcal{L}$. A time-index model is thus, still only a function of time-index features, while the learning algorithm is a function of past observations, i.e. $f, f_0 \in \mathcal{H}, f : \mathbb{R}^c \to \mathbb{R}^m, f = \mathcal{A}(f_0, \boldsymbol{Y}_{t-L:t})$. DeepTime as a forecaster, is a deep time-index model **endowed with a meta-learning algorithm**. In order to perform forecasting, it has to perform an inner loop optimization step defined by the learning algorithm, as highlighted in Equation (2). For the special case where we use the closed-form ridge regressor, the inner loop learning algorithm reduces to a form which is linear in the lookback window. Still, the deep time-index model is only a function of time-index features.

# E  DEEPTIME PSEUDOCODE

---

**Algorithm 1** PyTorch-Style Pseudocode of Closed-Form Ridge Regressor

---

mm: matrix multiplication, diagonal: returns the diagonal elements of a matrix, add_: in-place addition
linalg.solve computes the solution of a square system of linear equations with a unique solution.

```
# X: inputs, shape: (n_samples, n_dim)
# Y: targets, shape: (n_samples, n_out)
# lambd: scalar value representing the regularization coefficient

n_samples, n_dim = X.shape

# add a bias term by concatenating an all-ones vector
ones = torch.ones(n_samples, 1)
X = cat([X, ones], dim=-1)

if n_samples >= n_dim:
    # standard formulation
    A = mm(X.T, X)
    A.diagonal().add_(softplus(lambd))
    B = mm(X.T, Y)
    weights = linalg.solve(A, B)
else:
    # Woodbury formulation
    A = mm(X, X.T)
    A.diagonal().add_(softplus(lambd))
    weights = mm(X.T, linalg.solve(A, Y))
w, b = weights[:-1], weights[-1:]
return w, b
```

---

**Algorithm 2** PyTorch-Style Pseudocode of DeepTIMe

---

rearrange: einops style tensor operations
mm: matrix multiplication

```
# x: input time-series, shape: (lookback_len, multivariate_dim)
# lookback_len: scalar value representing the length of the lookback window
# horizon_len: scalar value representing the length of the forecast horizon
# inr: implicit neural representation

time_index = linspace(0, 1, lookback_len + horizon_len) # shape: (lookback_len + horizon_len)
time_index = rearrange(time_index, 't -> t 1') # shape: (lookback_len + horizon_len, 1)
time_reprs = inr(time_index) # shape: (lookback_len + horizon_len, hidden_dim)

lookback_reprs = time_reprs[:lookback_len]
horizon_reprs = time_reprs[-horizon_len:]
w, b = ridge_regressor(lookback_reprs, x)
# w.shape = (hidden_dim, multivariate_dim), b.shape = (1, multivariate_dim)
preds = mm(horizon_reprs, w) + b
return preds
```

---

# F SYNTHETIC DATA

The training set for each synthetic data experiment consists 1000 functions/tasks, while the test set contains 100 functions/tasks. We ensure that there is no overlap between the train and test sets.

**Linear**   Samples are generated from the function $y = ax + b$ for $x \in [-1, 1]$. This means that each function/task consists of 400 evenly spaced points between -1 and 1. The parameters of each function/task (i.e. $a, b$) are sampled from a normal distribution with mean 0 and standard deviation of 50, i.e. $a, b \sim \mathcal{N}(0, 50^2)$.

**Cubic**   Samples are generated from the function $y = ax^3 + bx^2 + cx + d$ for $x \in [-1, 1]$ for 400 points. Parameters of each task are sampled from a continuous uniform distribution with minimum value of -50 and maximum value of 50, i.e. $a, b, c, d \sim \mathcal{U}(-50, 50)$.

**Sums of sinusoids**   Sinusoids come from a fixed set of frequencies, generated by sampling $\omega \sim \mathcal{U}(0, 12\pi)$. We fix the size of this set to be five, i.e. $\Omega = \{\omega_1, \ldots, \omega_5\}$. Each function is then a sum of $J$ sinusoids, where $J \in \{1, 2, 3, 4, 5\}$ is randomly assigned. The function is thus $y = \sum_{j=1}^{J} A_j \sin(\omega_{r_j} x + \varphi_j)$ for $x \in [0, 1]$, where the amplitude and phase shifts are freely chosen via $A_j \sim \mathcal{U}(0.1, 5), \varphi_j \sim \mathcal{U}(0, \pi)$, but the frequency is decided by $r_j \in \{1, 2, 3, 4, 5\}$ to randomly select a frequency from the set $\Omega$.

# G DATASETS

**ETT**[2] Zhou et al. (2021) - Electricity Transformer Temperature provides measurements from an electricity transformer such as load and oil temperature. We use the *ETTm2* subset, consisting measurements at a 15 minutes frequency.

**ECL**[3] - Electricity Consuming Load provides measurements of electricity consumption for 321 households from 2012 to 2014. The data was collected at the 15 mintue level, but is aggregated hourly.

**Exchange**[4] Lai et al. (2018) - a collection of daily exchange rates with USD of eight countries (Australia, United Kingdom, Canada, Switzerland, China, Japan, New Zealand, and Singapore) from 1990 to 2016.

**Traffic**[5] - dataset from the California Department of Transportation providing the hourly road occupancy rates from 862 sensors in San Francisco Bay area freeways.

**Weather**[6] - provides measurements of 21 meteorological indicators such as air temperature, humidity, etc., every 10 minutes for the year of 2020 from the Weather Station of the Max Planck Biogeochemistry Institute in Jena, Germany.

**ILI**[7] - Influenza-like Illness measures the weekly ratio of patients seen with ILI and the total number of patients, obtained by the Centers for Disease Control and Prevention of the United States between 2002 and 2021.

---

[2] https://github.com/zhouhaoyi/ETDataset
[3] https://archive.ics.uci.edu/ml/datasets/ElectricityLoadDiagrams20112014
[4] https://github.com/laiguokun/multivariate-time-series-data
[5] https://pems.dot.ca.gov/
[6] https://www.bgc-jena.mpg.de/wetter/
[7] https://gis.cdc.gov/grasp/fluview/fluportaldashboard.html

## G.1 NON-STATIONARITY OF REAL-WORLD DATASETS

Table 7: Summary of real-world datasets, results of Chow test, and Augmented Dickey-Fuller (ADF) test. The statistical tests are performed on each dimension separately, since they are designed for univariate time-series. We report the number of dimensions which reject/fail to reject the null hypothesis, depending on which indicates non-stationarity. These are reported at significance levels of 0.1, 0.05, and 0.01. Larger values for the Chow test statistic indicate more non-stationarity, and larger (less negative) values for the ADF test statistic indicates more non-stationarity.

| Dataset | ETTm2 | ECL | Exchange | Traffic | Weather | ILI |
|---|---|---|---|---|---|---|
| Dimensions | 7 | 321 | 8 | 862 | 21 | 7 |
| Timesteps | 57,600 | 26,304 | 7,588 | 17,544 | 52,696 | 966 |
| Sampling Frequency | 15 minutes | 1 hour | 1 day | 1 hour | 10 minutes | 1 week |
| Chow Test: # dims reject null hypothesis @ 0.1 | 7 | 321 | 6 | 792 | 19 | 5 |
| Chow Test: # dims reject null hypothesis @ 0.05 | 7 | 320 | 5 | 771 | 19 | 5 |
| Chow Test: # dims reject null hypothesis @ 0.01 | 7 | 317 | 4 | 734 | 19 | 3 |
| Chow Test: average Chow test statistic | 49.13 | 23.55 | 21.25 | 16.93 | 25.39 | 2.77 |
| ADF Test: # dims fail to reject @ 0.1 | 2 | 139 | 8 | 807 | 9 | 0 |
| ADF Test: # dims fail to reject @ 0.05 | 2 | 120 | 8 | 739 | 7 | 0 |
| ADF Test: # dims fail to reject @ 0.01 | 2 | 103 | 8 | 657 | 6 | 0 |
| ADF Test: average ADF test statistic | -4.48 | -5.24 | -2.40 | -3.29 | -4.49 | -6.89 |

Real-world datasets used in long sequence time-series forecasting suffers from non-stationarity. We first verify this qualitatively by visualizing histogram values across some dimensions for each dataset in Figure 6. This simple visualization already gives us a strong confirmation on the distribution mismatch between the training and testing phases. We further verify this quantitatively via two statistical tests, the Chow test, and Augmented Dickey-Fuller (ADF) test. The Chow test is a test of whether the true coefficients in two linear regressions on different data sets are equal. Rejecting the null hypothesis of equality of regression coefficients in the two periods indicates that the train and test regions are generated from different distributions. The ADF test tests the null hypothesis that a unit root is present in a time series sample. Not rejecting the null hypothesis indicates that a unit root is present, and is thus non-stationary. Presented in Table 7 along with some dataset statistics, we report the results of both tests and the number of dimensions which meet the criteria for non-stationarity (rejecting the null hypothesis for Chow test, and not rejecting the null hypothesis for the ADF test) over various significance levels. We observe that the real-world datasets exhibit high levels of non-stationarity across dimensions based on both tests.

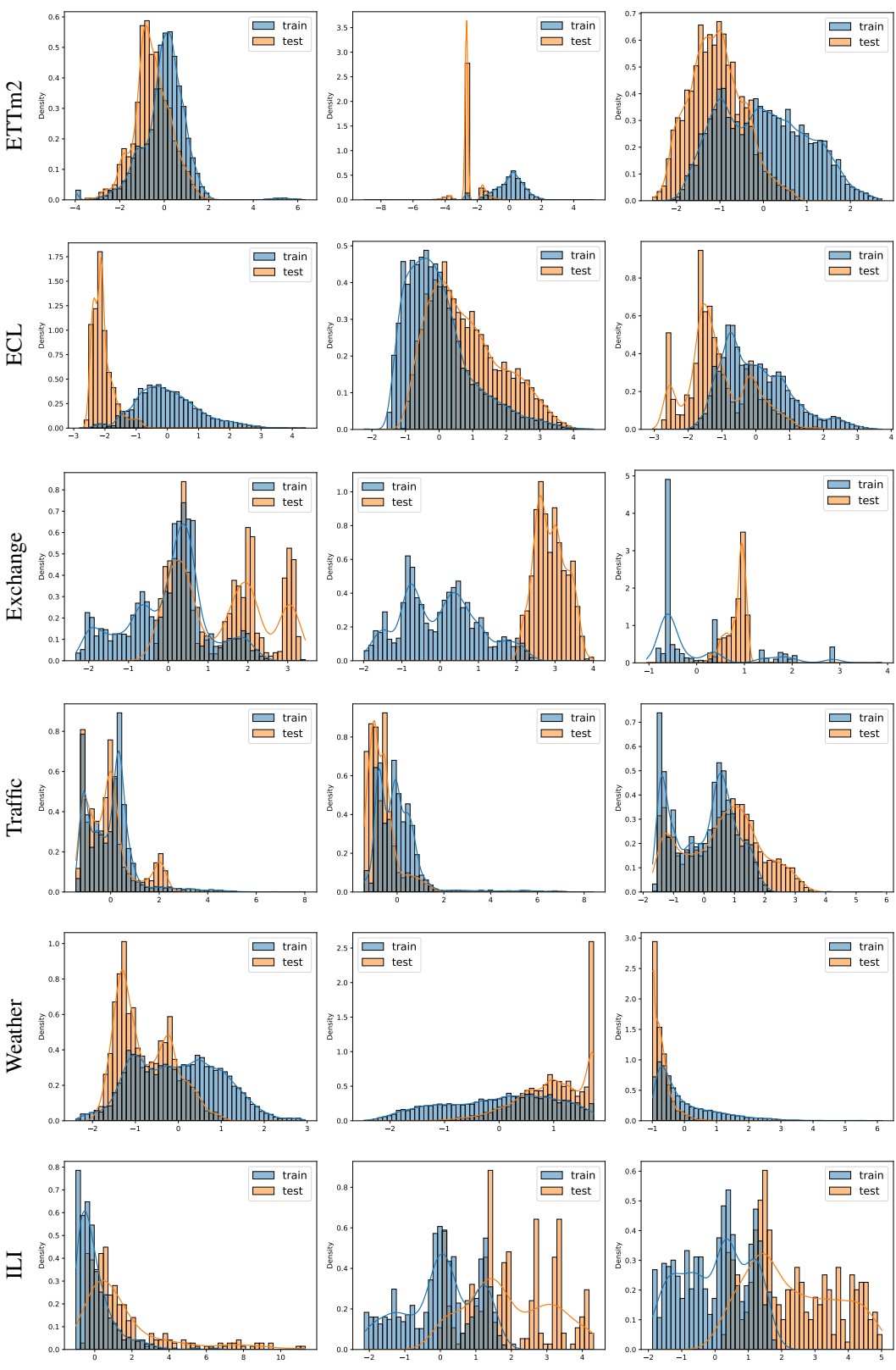

Figure 6: Histograms of time-series values from each real-world dataset presented in experiments. Visualized are the histograms of the train and test splits, verifying that even on the distribution of the marginals are different across time, signifying non-stationarity.

## H    DEEPTIME IMPLEMENTATION DETAILS

**Optimization**    We train DeepTime with the Adam optimizer (Kingma & Ba, 2014) with a learning rate scheduler following a linear warm up and cosine annealing scheme. Gradient clipping by norm is applied. The ridge regressor regularization coefficient, $\lambda$, is trained with a different, higher learning rate than the rest of the meta parameters. We use early stopping based on the validation loss, with a fixed patience hyperparameter (number of epochs for which loss deteriorates before stopping). All experiments are performed on an Nvidia A100 GPU.

**Model**    The ridge regression regularization coefficient is a learnable parameter constrained to positive values via a softplus function. We apply Dropout (Srivastava et al., 2014), then LayerNorm (Ba et al., 2016) after the ReLU activation function in each INR layer. The size of the random Fourier feature layer is set independently of the layer size, in which we define the total size of the random Fourier feature layer – the number of dimensions for each scale is divided equally.

## I    DEEPTIME HYPERPARAMETERS

Table 8: Hyperparameters used in DeepTime.

|  | Hyperparameter | Value |
|---|---|---|
| Optimization | Epochs | 50 |
|  | Learning rate | 1e-3 |
|  | $\lambda$ learning rate | 1.0 |
|  | Warm up epochs | 5 |
|  | Batch size | 256 |
|  | Early stopping patience | 7 |
|  | Max gradient norm | 10.0 |
| Model | Layers | 5 |
|  | Layer size | 256 |
|  | $\lambda$ initialization | 0.0 |
|  | Scales | $[0.01, 0.1, 1, 5, 10, 20, 50, 100]$ |
|  | Fourier features size | 4096 |
|  | Dropout | 0.1 |
|  | Lookback length multiplier, $\mu$ | $\mu \in \{1, 3, 5, 7, 9\}$ |

## J    UNIVARIATE FORECASTING BENCHMARK

Table 9: Univariate forecasting benchmark on long sequence time-series forecasting. Best results are highlighted in **bold**, and second best results are underlined.

| Methods | | DeepTime | | N-HiTS | | ETSformer | | Fedformer | | Autoformer | | Informer | | N-BEATS | | DeepAR | | Prophet | | ARIMA | | GP | |
|---|---|---|---|---|---|---|---|---|---|---|---|---|---|---|---|---|---|---|---|---|---|---|---|
| Metrics | | MSE | MAE | MSE | MAE | MSE | MAE | MSE | MAE | MSE | MAE | MSE | MAE | MSE | MAE | MSE | MAE | MSE | MAE | MSE | MAE | MSE | MAE |
| ETTm2 | 96 | 0.065 | **0.186** | 0.066 | **0.185** | 0.080 | 0.212 | **0.063** | 0.189 | 0.065 | 0.189 | 0.088 | 0.225 | 0.082 | 0.219 | 0.099 | 0.237 | 0.287 | 0.456 | 0.211 | 0.362 | 0.125 | 0.273 |
|  | 192 | 0.096 | 0.234 | **0.087** | **0.223** | 0.150 | 0.302 | 0.102 | 0.245 | 0.118 | 0.256 | 0.132 | 0.283 | 0.120 | 0.268 | 0.154 | 0.310 | 0.312 | 0.483 | 0.261 | 0.406 | 0.154 | 0.307 |
|  | 336 | 0.138 | 0.285 | **0.106** | **0.251** | 0.175 | 0.334 | 0.130 | 0.279 | 0.154 | 0.305 | 0.180 | 0.336 | 0.226 | 0.370 | 0.277 | 0.428 | 0.331 | 0.474 | 0.317 | 0.448 | 0.189 | 0.338 |
|  | 720 | 0.186 | 0.338 | **0.157** | **0.312** | 0.224 | 0.379 | 0.178 | 0.325 | 0.182 | 0.335 | 0.300 | 0.435 | 0.188 | 0.338 | 0.332 | 0.468 | 0.534 | 0.593 | 0.366 | 0.487 | 0.318 | 0.421 |
| Exchange | 96 | **0.086** | 0.226 | 0.093 | **0.223** | 0.099 | 0.230 | 0.131 | 0.284 | 0.241 | 0.299 | 0.591 | 0.615 | 0.156 | 0.299 | 0.417 | 0.515 | 0.828 | 0.762 | 0.112 | 0.245 | 0.165 | 0.311 |
|  | 192 | **0.173** | 0.330 | 0.230 | **0.313** | 0.223 | 0.353 | 0.277 | 0.420 | 0.273 | 0.665 | 1.183 | 0.912 | 0.669 | 0.665 | 0.813 | 0.735 | 0.909 | 0.974 | 0.304 | 0.404 | 0.649 | 0.617 |
|  | 336 | 0.539 | 0.575 | **0.370** | **0.486** | 0.421 | 0.497 | 0.426 | 0.511 | 0.508 | 0.605 | 1.367 | 0.984 | 0.611 | 0.605 | 1.331 | 0.962 | 1.304 | 0.988 | 0.736 | 0.598 | 0.596 | 0.592 |
|  | 720 | 0.936 | 0.763 | **0.728** | **0.569** | 1.114 | 0.807 | 1.162 | 0.832 | 0.991 | 0.860 | 1.872 | 1.072 | 1.111 | 0.860 | 1.890 | 1.181 | 3.238 | 1.566 | 1.871 | 0.935 | 1.002 | 0.786 |

# K  DEEPTIME STANDARD DEVIATION

Table 10: DeepTime main benchmark results with standard deviation. Experiments are performed over three runs.

(a) Multivariate benchmark.

| Metrics | | MSE (SD) | MAE (SD) |
|---|---|---|---|
| ETTm2 | 96 | 0.166 (0.000) | 0.257 (0.001) |
| | 192 | 0.225 (0.001) | 0.302 (0.003) |
| | 336 | 0.277 (0.002) | 0.336 (0.002) |
| | 720 | 0.383 (0.007) | 0.409 (0.006) |
| ECL | 96 | 0.137 (0.000) | 0.238 (0.000) |
| | 192 | 0.152 (0.000) | 0.252 (0.000) |
| | 336 | 0.166 (0.000) | 0.268 (0.000) |
| | 720 | 0.201 (0.000) | 0.302 (0.000) |
| Exchange | 96 | 0.081 (0.001) | 0.205 (0.002) |
| | 192 | 0.151 (0.002) | 0.284 (0.003) |
| | 336 | 0.314 (0.033) | 0.412 (0.020) |
| | 720 | 0.856 (0.202) | 0.663 (0.082) |
| Traffic | 96 | 0.390 (0.001) | 0.275 (0.001) |
| | 192 | 0.402 (0.000) | 0.278 (0.000) |
| | 336 | 0.415 (0.000) | 0.288 (0.001) |
| | 720 | 0.449 (0.000) | 0.307 (0.000) |
| Weather | 96 | 0.166 (0.001) | 0.221 (0.002) |
| | 192 | 0.207 (0.000) | 0.261 (0.000) |
| | 336 | 0.251 (0.000) | 0.298 (0.001) |
| | 720 | 0.301 (0.001) | 0.338 (0.001) |
| ILI | 24 | 2.425 (0.058) | 1.086 (0.027) |
| | 36 | 2.231 (0.087) | 1.008 (0.011) |
| | 48 | 2.230 (0.144) | 1.016 (0.037) |
| | 60 | 2.143 (0.032) | 0.985 (0.016) |

(b) Univariate benchmark.

| Metrics | | MSE (SD) | MAE (SD) |
|---|---|---|---|
| ETTm2 | 96 | 0.065 (0.000) | 0.186 (0.000) |
| | 192 | 0.096 (0.002) | 0.234 (0.003) |
| | 336 | 0.138 (0.001) | 0.285 (0.001) |
| | 720 | 0.186 (0.002) | 0.338 (0.002) |
| Exchange | 96 | 0.086 (0.000) | 0.226 (0.000) |
| | 192 | 0.173 (0.004) | 0.330 (0.003) |
| | 336 | 0.539 (0.066) | 0.575 (0.027) |
| | 720 | 0.936 (0.222) | 0.763 (0.075) |

## L  LOOKBACK LENGTH SENSITIVITY ANALYSIS

Table 11: Sensitivity analysis on the lookback window length. Results presented on the ETTm2 dataset across various values of the lookback length multiplier, $\mu$. Best results are highlighted in **bold**.

| Horizon | 96 | | 192 | | 336 | | 720 | |
|---|---|---|---|---|---|---|---|---|
| $\mu$ | MSE | MAE | MSE | MAE | MSE | MAE | MSE | MAE |
| 1 | 0.192 | 0.287 | 0.255 | 0.332 | 0.294 | 0.354 | 0.383 | 0.409 |
| 3 | 0.172 | 0.264 | 0.228 | 0.304 | 0.277 | **0.336** | **0.371** | **0.403** |
| 5 | 0.168 | 0.259 | 0.225 | 0.302 | **0.275** | 0.337 | 0.389 | 0.420 |
| 7 | 0.166 | **0.257** | **0.223** | **0.300** | 0.279 | 0.343 | 0.440 | 0.451 |
| 9 | **0.165** | 0.258 | **0.223** | 0.301 | 0.285 | 0.350 | 0.409 | 0.434 |

## M  ADDITIONAL ABLATION STUDY

Table 12: Additional ablation study on variants of DeepTime. + *Finetune* refers to training an INR via gradient descent for each lookback window on top of having a training phase. *Full MAML* refers to performing the full meta-learning formulation on the whole model rather than just the last layer, using gradient-based optimization.

| Methods | | DeepTime | | + Finetune | | + Finetune + Datetime | | Full MAML | |
|---|---|---|---|---|---|---|---|---|---|
| Metrics | | MSE | MAE | MSE | MAE | MSE | MAE | MSE | MAE |
| ETTm2 | 96 | **0.166** | **0.257** | 3.028 | 1.328 | 3.242 | 1.365 | 0.235 | 0.326 |
| | 192 | **0.225** | **0.302** | 3.043 | 1.341 | 3.385 | 1.391 | 0.295 | 0.361 |
| | 336 | **0.277** | **0.336** | 2.950 | 1.331 | 3.367 | 1.387 | 0.348 | 0.392 |
| | 720 | **0.383** | **0.409** | 2.721 | 1.253 | 3.476 | 1.407 | 0.491 | 0.484 |

## N  RANDOM FOURIER FEATURES SCALE HYPERPARAMETER SENSITIVITY ANALYSIS

Table 13: Results from hyperparameter sweep on the scale hyperparameter. Best scores are highlighted in **bold**, and worst scores are highlighted in **bold red**.

| Scale Hyperparam | | 0.01 | | 0.1 | | 1 | | 5 | | 10 | | 20 | | 50 | | 100 | |
|---|---|---|---|---|---|---|---|---|---|---|---|---|---|---|---|---|---|
| Metrics | | MSE | MAE | MSE | MAE | MSE | MAE | MSE | MAE | MSE | MAE | MSE | MAE | MSE | MAE | MSE | MAE |
| ETTm2 | 96 | **0.216** | **0.300** | 0.189 | 0.285 | 0.173 | 0.268 | 0.168 | 0.262 | 0.166 | 0.260 | 0.165 | 0.258 | 0.165 | 0.259 | **0.164** | **0.257** |
| | 192 | **0.275** | **0.340** | 0.264 | 0.333 | 0.239 | 0.317 | 0.225 | **0.301** | 0.225 | 0.303 | 0.224 | 0.302 | 0.224 | 0.304 | **0.220** | 0.301 |
| | 336 | **0.340** | **0.375** | 0.319 | 0.371 | 0.292 | 0.351 | **0.275** | 0.337 | 0.277 | **0.336** | 0.282 | 0.345 | 0.278 | 0.342 | 0.280 | 0.344 |
| | 720 | **0.424** | 0.430 | 0.405 | 0.420 | 0.381 | 0.412 | **0.364** | **0.392** | 0.375 | 0.408 | 0.410 | **0.430** | 0.396 | 0.423 | 0.406 | 0.429 |

## O  ABLATION STUDIES DETAILS

In this section, we list more details on the models compared to in the ablation studies section. Unless otherwise stated, we perform the same hyperparameter tuning for all models in the ablation studies, and use the same standard hyperparameters such as number of layers, layer size, etc.

### O.1  ABLATION STUDY ON VARIANTS OF DEEPTIME

**RR**  Removing the ridge regressor module refers to replacing it with a simple linear layer, $\text{Linear} : \mathbb{R}^d \to \mathbb{R}^m$, where $\text{Linear}(\boldsymbol{x}) = \boldsymbol{W}\boldsymbol{x} + \boldsymbol{b}, \boldsymbol{x} \in \mathbb{R}^d, \boldsymbol{W} \in \mathbb{R}^{m \times d}, \boldsymbol{b} \in \mathbb{R}^m$. This corresponds to a straight forward INR, which is trained across all lookback-horizon pairs in the dataset.

**Local**  For models marked "Local", we similarly remove the ridge regressor module and replace it with a linear layer. Yet, the model is not trained across all lookback-horizon pairs in the dataset.

Instead, for each lookback-horizon pair in the validation/test set, we fit the model to the lookback window via gradient descent, and then perform prediction on the horizon to obtain the forecasts. A new model is trained from scratch for each lookback-horizon window. We perform tuning on an extra hyperparameter, the number of epochs to perform gradient descent, for which we search through $\{10, 20, 30, 40, 50\}$.

**Datetime Features**  As each dataset comes with a timestamps for each observation, we are able to construct datetime features from these timestamps. We construct the following features:

1. Quarter-of-year
2. Month-of-year
3. Week-of-year
4. Day-of-year
5. Day-of-month
6. Day-of-week
7. Hour-of-day
8. Minute-of-hour
9. Second-of-minute

Each feature is initially an integer value, e.g. month-of-year can take on values in $\{0, 1, \ldots, 11\}$, which we subsequently normalize to a $[0, 1]$ range. Depending on the data sampling frequency, the appropriate features can be chosen. For the ETTm2 dataset, we used all features except second-of-minute since it is sampled at a 15 minute frequency.

## O.2  Ablation study on backbone models

For all models in this section, we retain the differentiable closed-form ridge regressor, to identify the effects of the backbone model used.

**MLP**  The random Fourier features layer is a mapping from coordinate space to latent space $\gamma : \mathbb{R}^c \to \mathbb{R}^d$. To remove the effects of the random Fourier features layer, we simply replace it with a with a linear map, $\text{Linear} : \mathbb{R}^c \to \mathbb{R}^d$.

**SIREN**  We replace the random Fourier features backbone with the SIREN model which is introduced by Sitzmann et al. (2020b). In this model, periodical activation functions are used, i.e. $\sin(\boldsymbol{x})$, along with specified weight initialization scheme.

**RNN**  We use a 2 layer LSTM with hidden size of 256. Inputs are observations, $\boldsymbol{y}_t$, in an IMS fashion, predicting the next time step, $\boldsymbol{y}_{t+1}$.

## P  Computational Efficiency Experiments Details

**Trans/In/Auto/ETS-former**  We use a model with 2 encoder and 2 decoder layers with a hidden size of 512, as specified in their original papers.

**N-BEATS**  We use an N-BEATS model with 3 stacks and 3 layers (relatively small compared to 30 stacks and 4 layers used in their orignal paper[8]), with a hidden size of 512. Note, N-BEATS is a univariate model and values presented here are multiplied by a factor of $m$ to account for the multivariate data. Another dimension of comparison is the number of parameters used in the model. Demonstrated in Table 14, fully connected models like N-BEATS, their number of parameters scales linearly with lookback window and forecast horizon length, while for Transformer-based and Deep-Time, the number of parameters remains constant.

---

[8]`https://github.com/ElementAI/N-BEATS/blob/master/experiments/electricity/generic.gin`

**N-HiTS**   We use an N-HiTS model with hyperparameters as sugggested in their original paper (3 stacks, 1 block in each stack, 2 MLP layers, 512 hidden size). For the following hyperparameters which were not specified (subject to hyperparameter tuning), we set the pooling kernel size to $[2, 2, 2]$, and the number of stack coefficients to $[24, 12, 1]$. Similar to N-BEATS, N-HiTS is a univariate model, and values were multiplied by a factor of $m$ to account for the multivariate data.

Table 14: Number of parameters in each model across various lookback window and forecast horizon lengths. The models were instantiated for the ETTm2 multivariate dataset (this affects the embedding and projection layers in Autoformer. Values for N-HiTS in this table are **not** multiplied by $m$ since it is a global model (i.e. a single univariate model is used for all dimensions of the time-series).

| Methods | | Autoformer | N-HiTS | DeepTime |
|---|---|---|---|---|
| Lookback | 48 | 10,535,943 | 927,942 | 1,314,561 |
| | 96 | 10,535,943 | 1,038,678 | 1,314,561 |
| | 168 | 10,535,943 | 1,204,782 | 1,314,561 |
| | 336 | 10,535,943 | 1,592,358 | 1,314,561 |
| | 720 | 10,535,943 | 2,478,246 | 1,314,561 |
| | 1440 | 10,535,943 | 4,139,286 | 1,314,561 |
| | 2880 | 10,535,943 | 7,461,366 | 1,314,561 |
| | 5760 | 10,535,943 | 14,105,526 | 1,314,561 |
| Horizon | 48 | 10,535,943 | 927,942 | 1,314,561 |
| | 96 | 10,535,943 | 955,644 | 1,314,561 |
| | 168 | 10,535,943 | 997,197 | 1,314,561 |
| | 336 | 10,535,943 | 1,094,154 | 1,314,561 |
| | 720 | 10,535,943 | 1,315,770 | 1,314,561 |
| | 1440 | 10,535,943 | 1,731,300 | 1,314,561 |
| | 2880 | 10,535,943 | 2,562,360 | 1,314,561 |
| | 5760 | 10,535,943 | 4,224,480 | 1,314,561 |

## Q  GENERALIZATION BOUND FOR OUR META-LEARNING FRAMEWORK

In this section, we derive a meta-learning generalization bound for DeepTime under the PAC-Bayes framework (Shalev-Shwartz & Ben-David, 2014). Our formulation follows (Amit & Meir, 2018) and assumes that all tasks share the same hypothesis space $\mathcal{H}$, sample space $\mathcal{Z}$ and loss function $\ell : \mathcal{H} \times \mathcal{Z} \to [0, 1]$. We observes $n$ tasks in the form of sample sets $\mathcal{D}_1, \ldots, \mathcal{D}_n$. The number of samples in each task is $H + L$. Each dataset $\mathcal{D}_k$ is assumed to be generated *i.i.d* from an unknown sample distribution $D_k^{H+L}$. Each task's sample distribution $D_k$ is *i.i.d.* generated from an unknown meta distribution, $E$. Particularly, we have $\mathcal{D}_k = (z_{k-L}, \ldots, z_k, \ldots, z_{k+H-1})$, where $z_t = (\tau_t, \boldsymbol{y}_t)$. Here, $\tau_t$ is the time coordinate, and $\boldsymbol{y}_t$ is the time-series value. For any forecaster $h(\cdot)$ parameterized by $\theta$, we define the loss function $\ell(h_\theta, z_t)$. We also define $P$ as the prior distribution over $\mathcal{H}$ and $Q$ as the posterior over $\mathcal{H}$ for each task. In the meta-learning setting, we assume a hyper-prior $\mathcal{P}$, which is a prior distribution over priors, observes a sequence of training tasks, and then outputs a distribution over priors, called hyper-posterior $\mathcal{Q}$.

**Theorem Q.1.** *Consider the Meta-Learning framework, given the hyper-prior $\mathcal{P}$, then for any hyper-posterior $\mathcal{Q}$, any $c_1, c_2 > 0$ and any $\delta \in (0, 1]$ with probability $\geq 1 - \delta$ we have,*

$$P\left(er(\mathcal{Q}) \leq \frac{c_1 c_2}{(1 - e^{-c_1})(1 - e^{-c_2})} \cdot \frac{1}{n}\sum_{k=1}^{n} \hat{er}(\mathcal{Q}, \mathcal{D}_k) + \frac{c_1}{1 - e^{-c_1}} \cdot \frac{\mathrm{KL}(\mathcal{P}||\mathcal{Q}) + \log\frac{1}{\delta_k}}{nc_1} \right.$$
$$\left. + \frac{c_1 c_2}{(1 - e^{-c_2})(1 - e^{-c_1})} \cdot \frac{\mathrm{KL}(\rho||\pi) + \log\frac{1}{\delta_k}}{n(H+L)c_2}\right) \geq 1 - \delta. \quad (5)$$

*Proof.* Our proof contains two steps. First, we bound the error within observed tasks due to observing a limited number of samples. Then we bound the error on the task environment level due to observing a finite number of tasks. Both of the two steps utilize Catoni's classical PAC-Bayes bound (Catoni, 2007) to measure the error. We give here the Catoni's classical PAC-Bayes bound.

**Theorem Q.2.** *(Catoni's bound (Catoni, 2007)) Let $\mathcal{X}$ be a sample space, $P(X)$ a distribution over $\mathcal{X}$, $\Theta$ a hypothesis space. Given a loss function $\ell(\theta, X) : \Theta \times \mathcal{X} \to [0, 1]$ and a collection of M i.i.d random variables $(X_1, \ldots, X_M)$ sampled from $P(X)$. Let $\pi$ be a prior distribution over hypothesis space. Then, for any $\delta \in (0, 1]$ and any real number $c > 0$, the following bound holds uniformly for all posterior distributions $\rho$ over hypothesis space,*

$$P\left(\mathop{\mathbb{E}}_{X_i \sim P(X), \theta \sim \rho}[\ell(\theta, X_i)] \leq \frac{c}{1 - e^{-c}}\left[\frac{1}{M}\sum_{m=1}^{M} \mathop{\mathbb{E}}_{\theta \sim \rho}[\ell(\theta, X_m)] + \frac{\mathrm{KL}(\rho||\pi) + \log\frac{1}{\delta}}{Mc}\right], \forall \rho\right)$$
$$\geq 1 - \delta.$$

We first utilize Theorem Q.2 to bound the generalization error in each of the observed tasks. Let $k$ be the index of task, we have the definition of expected error and empirical error as follows,

$$er(\mathcal{Q}, D_k) = \mathop{\mathbb{E}}_{P \sim \mathcal{Q}} \mathop{\mathbb{E}}_{h \sim Q(\mathcal{D}_k, P)} \mathop{\mathbb{E}}_{z \sim D_k} \ell(h, z), \quad (6)$$

$$\hat{er}(\mathcal{Q}, \mathcal{D}_k) = \mathop{\mathbb{E}}_{P \sim \mathcal{Q}} \mathop{\mathbb{E}}_{h \sim Q(\mathcal{D}_k, P)} \frac{1}{H+L}\sum_{j=k-L}^{k+H-1} \ell(h, z_j). \quad (7)$$

Then, according to Theorem Q.2, for any $\delta_k \sim (0, 1]$ and $c_2 > 0$, we have

$$P\left(er(\mathcal{Q}, D_k) \leq \frac{c_2}{1 - e^{-c_2}}\hat{er}(\mathcal{Q}, \mathcal{D}_k) + \frac{c_2}{1 - e^{-c_2}} \cdot \frac{\mathrm{KL}(\rho||\pi) + \log\frac{1}{\delta_k}}{(H+L)c_2}\right) \geq 1 - \delta_k. \quad (8)$$

Next, we bound the error due to observing a limited number of tasks from the environment. Similarly, we have the definition of expected task error as follows

$$er(\mathcal{Q}) = \mathop{\mathbb{E}}_{D \sim E} \mathop{\mathbb{E}}_{\mathcal{D} \sim D^{H+L}} \mathop{\mathbb{E}}_{P \sim \mathcal{Q}} \mathop{\mathbb{E}}_{h \sim Q(\mathcal{D}, P)} \mathop{\mathbb{E}}_{z \sim D} \ell(h, z)$$
$$= \mathop{\mathbb{E}}_{D \sim E} \mathop{\mathbb{E}}_{\mathcal{D} \sim D^{H+L}} er(\mathcal{Q}, D). \quad (9)$$

Then we have the definition of error across the $n$ tasks,

$$\frac{1}{n} \sum_{k=1}^{n} \mathop{\mathbb{E}}_{P \sim \mathcal{Q}} \mathop{\mathbb{E}}_{h \sim Q(\mathcal{D}_k, P)} \mathop{\mathbb{E}}_{z \sim D_k} \ell(h, z) = \frac{1}{n} \sum_{k=1}^{n} er(\mathcal{Q}, D_k). \tag{10}$$

Then Theorem Q.2 says that the following holds for any $\delta_0 \sim (0, 1]$ and $c_1 > 0$, we have

$$P \left( er(\mathcal{Q}) \leq \frac{c_1}{1 - e^{-c_1}} \frac{1}{n} \sum_{k=1}^{n} er(\mathcal{Q}, D_k) + \frac{c_1}{1 - e^{-c_1}} \cdot \frac{\mathrm{KL}(\mathcal{P}||\mathcal{Q}) + \log \frac{1}{\delta_k}}{n c_1} \right) \geq 1 - \delta_0. \tag{11}$$

Finally, by employing the union bound, we could bound the probability of the intersection of the events in Equation (11) and Equation (8) For any $\delta > 0$, set $\delta_0 = \frac{\delta}{2}$ and $\delta_i = \frac{\delta}{2n}$ for $i = 1, \ldots, n$,

$$P \left( er(\mathcal{Q}) \leq \frac{c_1 c_2}{(1 - e^{-c_1})(1 - e^{-c_2})} \cdot \frac{1}{n} \sum_{k=1}^{n} \hat{er}(\mathcal{Q}, \mathcal{D}_k) + \frac{c_1}{1 - e^{-c_1}} \cdot \frac{\mathrm{KL}(\mathcal{P}||\mathcal{Q}) + \log \frac{1}{\delta_k}}{n c_1} \right.$$

$$\left. + \frac{c_1 c_2}{(1 - e^{-c_2})(1 - e^{-c_1})} \cdot \frac{\mathrm{KL}(\rho||\pi) + \log \frac{1}{\delta_k}}{n(H + L)c_2} \right) \geq 1 - \delta. \tag{12}$$

$\square$

Theorem Q.1 shows that the expected task generalization error is bounded by the empirical multi-task error plus two complexity terms. The first term represents the complexity of the environment, or equivalently, the time-series dataset, converging to zero if we observe an infinitely long time-series ($n \to \infty$). The second term represents the complexity of the observed tasks, or equivalently, the lookback-horizon windows. This converges to zero when there are sufficient number of time steps in each window ($H + L \to \infty$).

