# OpenReview forum: "DeepTime: Deep Time-index Meta-learning for Non-stationary Time-series Forecasting"
_ICLR.cc/2023/Conference — Submitted to ICLR 2023_

### Official Review · Reviewer_comQ · 2022-10-24

**Confidence:** 4
**Correctness:** 2
**Technical Novelty And Significance:** 3
**Empirical Novelty And Significance:** 2
**Recommendation:** 3

**Clarity, Quality, Novelty And Reproducibility:**

The paper clarity is okay. The methods section could be improved. I suggest including the pseudocode for training and inference.

E.g. for INR, "We make use a of them as they are a natural fit for time-index models," - such motivations are not well supported.

The quality is mediocre. Some key aspects have issues as mentioned.

The idea is novel in time-series.

Reproducibility is concerning, as not all details are included.

**Strength And Weaknesses:**

Strengths

- Forecasting as meta learning framework is an interesting and canonical idea.

- Extensive ablation analyses

Weaknesses

- There is not sufficient explanation on what non-stationarity is.

- The method is proposed for non-stationary time-series, but the experiments are not specifically obtained for non-stationary time-series data. The synthetic datasets are simple functions and most of the real-world benchmarks are not from applications where non-stationary dynamics are prominent, maybe except Exchange, on which the results are weaker.

- It is unclear why Fourier features would be a good idea for non-stationary time series, as opposed to time-varying frequency representations.

- There is no analysis on how much different datasets contain covariate or distribution shifts.

- Some relevant baselines are skipped:
https://arxiv.org/abs/2205.14415
https://arxiv.org/abs/2205.13504
https://arxiv.org/abs/1912.09363

- For computational efficiency, the impact of parallelization is ignored.

- The impact of lookback length is not sufficiently analyzed.

- For a non-stationary dataset, training and validation distributions would have differences. How would it affect the hparam selection? This aspect is ignored.

**Summary Of The Paper:**

The paper proposes DeepTime, a deep time-index model trained via meta-learning, with the motivations for stronger smoothness prior, avoiding the problem of covariate shift, and having better sample efficiency. Experimentation on various time series benchmarks are shown.

**Summary Of The Review:**

Because of the issues above, mainly lack of convincing empirical validation appropriate for non-stationarity, the paper cannot be accepted in its current form in my opinion.

---

> ### Author Response · Authors · 2022-11-18
> **Response to Reviewer comQ (2/2)**
>
> __Q7: For a non-stationary dataset, training and validation distributions would have differences. How would it affect the hyperparameter selection? This aspect is ignored.__
>
> A7: Hyperparameter selection procedure should not be affected by a shift in distribution between training and validation sets. The goal is ultimately to select a set of hyperparameters which perform well on the validation distribution, and to hope that it performs well on the test set as well. If the model generalizes well (or in our case, meta-learns well), then a standard hyperparameter selection procedure would result in good test set results too.
>
> Could the reviewer give more specific questions or comments regarding this point if they still have doubts?
>
> __Q8: The methods section could be improved. I suggest including the pseudocode for training and inference.__
>
> A8: Thank you for the suggestion, pseudocode is available in Appendix E, which we link to in Section 2, paragraph 2.
>
> __Q9: Reproducibility is concerning, as not all details are included.__
>
> A9: We respectfully disagree with this point as we have included as much details for reproducibility as possible. We urge the reviewer to see Section 3.2, and Appendices G, H, I, K, O, P for more experimental details. Furthermore, we have also included our code for maximum reproducibility, and also elaborated on the hyperparameter search range, as well as the exact hyperparameter values resulting from hyperparameter tuning.
>
> Baselines have also been extensively tuned by their authors, as described in their respective papers [1,2,3,4].
>
> Finally, if the reviewer feels that there are any missing information for reproducibility, please do notify us of any specific details that should be included, and we will do our utmost best to include them.
>
> __Q10: Some relevant baslines are skipped [5,6,7].__
>
> A10: Thank you for the pointers to these relevant works. We have included Non-stationary Transformers [5], recently published at NeurIPS 2022 as a baseline, while [6] is a preprint yet to be published. [7] is an old work, superseded by many recent Transformer based forecasters [1,3,4], and is also not a related baseline since it is neither tackling non-stationary forecasting nor is it tackling long sequence time-series forecasting.
>
> ---
> [1] Zhou, H., Zhang, S., Peng, J., Zhang, S., Li, J., Xiong, H., & Zhang, W. (2021, May). Informer: Beyond efficient transformer for long sequence time-series forecasting. In Proceedings of the AAAI Conference on Artificial Intelligence (Vol. 35, No. 12, pp. 11106-11115).
>
> [2] Challu, C., Olivares, K. G., Oreshkin, B. N., Garza, F., Mergenthaler, M., & Dubrawski, A. (2022). N-hits: Neural hierarchical interpolation for time series forecasting. arXiv preprint arXiv:2201.12886.
>
> [3] Wu, H., Xu, J., Wang, J., & Long, M. (2021). Autoformer: Decomposition transformers with auto-correlation for long-term series forecasting. Advances in Neural Information Processing Systems, 34, 22419-22430.
>
> [4] Zhou, T., Ma, Z., Wen, Q., Wang, X., Sun, L. &amp; Jin, R.. (2022). FEDformer: Frequency Enhanced Decomposed Transformer for Long-term Series Forecasting. <i>Proceedings of the 39th International Conference on Machine Learning</i>, in <i>Proceedings of Machine Learning Research</i> 162:27268-27286
>
> [5] Yong Liu, Haixu Wu, Jianmin Wang, and Mingsheng Long. Non-stationary transformers: Exploring the stationarity in time series forecasting. In Alice H. Oh, Agarwal, Danielle Belgrave, and Kyunghyun Cho (eds.), Advances in Neural Information Processing Systems, 2022. URL https://openreview.net/forum?id=ucNDIDRNjjv.
>
> [6] Zeng, A., Chen, M., Zhang, L., & Xu, Q. (2022). Are Transformers Effective for Time Series Forecasting?. arXiv preprint arXiv:2205.13504.
>
> [7] Lim, B., Arık, S. Ö., Loeff, N., & Pfister, T. (2021). Temporal fusion transformers for interpretable multi-horizon time series forecasting. International Journal of Forecasting, 37(4), 1748-1764.

---

> ### Author Response · Authors · 2022-11-18
> **Response to Reviewer comQ (1/2)**
>
> Thank you Reviewer comQ for taking the time to read and review our work. Overall, we hope that our answers to the specific questions raised clears any doubts the reviewer might have with our work. We hope to continue engaging the reviewer throughout the discussion period if any further doubts persists, specifically, to get more details regarding the reviewer's concerns on reproducibility, and other issues.
>
> __Q1: There is insufficient explanation on what non-stationarity is.__
>
> A1: Thank you for raising this important point, please see our response to all reviewers for detailed definition on the problems raised by non-stationarity, i.e. covariate shift and conditional distribution shift.
>
> __Q2: The experiments are not performed specifically on non-stationary time-series data.__
>
> A2: Thank you for raising this important point, please see our response to all reviewers and Appendix G.1 for a detailed analysis on the stationarity of the real-world datasets. We further highlight that one of the additional baselines [5] (NeurIPS'22) the reviewer has requested, also tackles non-stationarity on the same real-world datasets.
>
> __Q3: Why are Implicit Neural Representations a natural fit for time-index models?__
>
> A3: Implicit Neural Representations, also known as Coordinate-Based Models, are defined simply by $y_c = f(c)$, where $f$ is parameterized as a multi-layer perceptron. They have been used on 2d data where $c \in \mathbb{R}^2$ (images, mapping pixel coordinates to pixel values), and more popularly, in neural graphics to represent objects such as point clouds.
>
> Time-series is the special case where $c \in \mathbb{R}$, where the coordinates are 1d time-index. A deep time-index model would be defined similarly to an INR, to map the time-index to the value at that time step, using a multi-layered perceptron. Thus, we connect the literature of INRs, to time-index models.
>
> __Q4: It is unclear why Fourier features would be a good idea for non-stationary time-series, as opposed to time-varying frequency representations.__
>
> A4: We highlight that Fourier features are ___not proposed for the sake of tackling non-stationary time-series___. Instead, it has been established in the INR literature, that using simple ReLU MLPs to map a coordinate to a value suffers from "spectral bias", whereby attempting to reconstruct an image leads to a smoothed version of the image. Thus, to overcome this, they introduce random Fourier features, whereby in-depth studies have been shown to overcome this spectral bias.
>
> This phenomena is experienced for all kinds of data, including 1d data such as time-series. As such, we also include the same module for an effective time-index model. Furthermore, we overcome the drawback of extensive hyperparameter search with our novel concatenated Fourier features.
>
> __Q5: For computational efficiency, the impact of parallelization is ignored.__
>
> A5: All computational efficiency experiments use the same hardware as mentioned in Appendix H, using the same GPU acceleration across models for fair comparison. Thus, the impact of parallelization has not been ignored. Please let us know if this addresses your concerns, if not, we kindly request that the reviewer elaborate on why the impact of parallelization is ignored.
>
> __Q6: The impact of lookback length is not sufficiently analyzed.__
>
> A6: In the real-world data experiments, as per the established convention, the size of the lookback window, $L$, is treated as a hyperparameter. As specified in Section 3.2 and Appendix I, we search through values of $L$ as a multiple of the horizon length $H$, by setting $L = \mu * H$, for $\mu = \{1, 3, 5, 7, 9\}$. Benchmark baselines similarly tune lookback length as a hyperparameter [1,2].
>
> Below is a sensitivity analysis on the value of $\mu$ across various horizons on the ETTm2 dataset. We observe that for shorter horizons, a larger value of $\mu$ is preferred while for longer horizons, a smaller value of $\mu$ is preferred, indicating an optimal lookback length which is similar regardless of horizon length.
>
> | Horizon |     96    |           |    192    |           |    336    |           |    720    |           |
> |:-------:|:---------:|:---------:|:---------:|:---------:|:---------:|:---------:|:---------:|:---------:|
> |  $\mu$  |    MSE    |    MAE    |    MSE    |    MAE    |    MSE    |    MAE    |    MSE    |    MAE    |
> |    1    |   0.192   |   0.287   |   0.255   |   0.332   |   0.294   |   0.354   | **0.383** | **0.409** |
> |    3    |   0.172   |   0.264   |   0.228   |   0.304   |   0.277   | **0.336** |   0.371   |   0.403   |
> |    5    |   0.168   |   0.259   |   0.225   |   0.302   | **0.275** |   0.337   |   0.389   |   0.420   |
> |    7    |   0.166   |   0.257   | **0.223** | **0.300** |   0.279   |   0.343   |   0.440   |   0.451   |
> |    9    | **0.165** | **0.258** | **0.223** |   0.301   |   0.285   |   0.350   |   0.409   |   0.434   |

---

> ### Author Response · Authors · 2022-11-25
> **Further Discussions**
>
> Dear Reviewer comQ,
>
> We hope that you have had the time to go through our response and revised manuscript. This is a gentle reminder and request, if our response has been satisfactory, that you consider raising your recommendation. If not, we look forward to further discussions with you.
>
> Best regards,
>
> Authors

---

> ### Author Response · Authors · 2022-12-05
> **Gentle reminder**
>
> Dear Reviewer comQ,
>
> Hope you are well! As the discussion period is coming to an end soon, we are looking forward to any feedback you have to our response despite your busy schedule. We will highly appreciate it if you let us know whether your previous concerns have been adequately addressed. Thank you once again.
>
> We have done our utmost to address the concerns as you suggested:
>
> 1. We __included a formal definition and presentation__ on covariate and conditional distribution shift, and included an __extensive empirical analysis__ to verify the non-stationarity of real-world datasets
> 2. We  __clarified the possible misunderstandings__, concerning our proposed use Implicit Neural Representations, and Fourier features.
> 3. We __included additional empirical analysis__ on the impact of lookback window length.
> 4. We __addressed possible misunderstandings__ regarding the reproducibility of our work.
>
> Thank you again for your dedication to reviewing our paper and we are looking forward to your feedback.

---

> ### Author Response · Authors · 2022-12-07
> **Gentle reminder for further comments**
>
> Dear Reviewer comQ,
>
> Hope everything is going well. As the discussion period is coming to an end very soon, we would like to send a gentle reminder, and to let you know that we look forward to hearing any further updates and thoughts you may have. Once again, thank you for your efforts and service.

---

### Official Review · Reviewer_3pqc · 2022-10-24

**Confidence:** 4
**Correctness:** 3
**Technical Novelty And Significance:** 3
**Empirical Novelty And Significance:** 3
**Recommendation:** 6

**Clarity, Quality, Novelty And Reproducibility:**

On the whole, the paper was clearly presented and the motivations were very intuitive as well. A few questions remain, however, which would be very helpful with reproducibility.

Questions
---
1.	Is any feature engineering applied to inputs for benchmarks?
2.	How are lookback windows determined?
3.	How much data was used for training in each case?


**Strength And Weaknesses:**

Strengths
---

As neural networks notoriously underperform in the presence of covariate shifts – which are common in non-stationary time series datasets – the creation of methods that allow neural networks to adapt to changing regimes is extremely important. The use of meta-learning here is an interesting approach, and the out-performance of other state-of-the-art transformers makes it a promising method.

Weaknesses
---

However, I do have several concerns regarding experimental evaluation, in particular:

1.	Lack of hyperparameter optimisation – while the paper does show out-performance across benchmarks, we note that all the hyperparameters of DeepTime and benchmarks have been pre-specified based on “suggested” settings. Given that optimal hyperparameters can vary largely across time-series datasets – for example smaller models in data-limited regimes vs larger models where data is abundant – the lack of hyperparameter optimisation does beg the question of whether the performance differences here are due to improperly selected hyperparams and not model improvements.

2.	Are improvements due to differences in features used? – While the use of random Fourier features helps to make the time index range bound (which would otherwise introduce its own train-test mismatch), the final output of the INR would essentially be a set of non-linear seasonal features which repeat over time. Out-performance hence could be due to these seasonal relationships being more prevalent in the datasets used. As such, it would be good to see 1) how properly tuned transformer/tabular models perform when given the same random Fourier features + observations, and 2) whether a rolling linear regression on random Fourier features is sufficient for performance.

3.	Does the meta learner handle non-stationarity? – while promising in theory, it is unclear whether there are in fact concept drifts in the time series datasets used for benchmarking. Tests on simulated data that incorporates regime changes over time could be useful here, with performance changes/decays across regimes documented for all models.


**Summary Of The Paper:**

The paper proposes a new deep learning model for forecasting under non-stationary conditions – utilising a network that builds global representations of random Fourier features and projects it using meta-learning framework that recalibrates the output layer with each window.

**Summary Of The Review:**

I do think the idea and intuition behind the model is compelling, but would require improvements in experiments and benchmarks to be more convincing for me.

---

> ### Author Response · Authors · 2022-11-18
> **Response to Reviewer 3pqc (2/2)**
>
> __Q3: Does the meta learner handle non-stationarity? It is unclear whether there are in fact concept drifts in the time series datasets used for benchmarking.__
>
> A3: Thank you for raising up this important issue, please see our response to all reviewers regarding this, where we have performed qualitative and quantitative analyses on the non-stationarity of real-world datasets used for the experiments.
>
> __Q4: Is any feature engineering applied to inputs for benchmarks?__
>
> A4: In general, extensive feature engineering has not been used in the prior work of long sequence time-series forecasting. Input normalization is applied by subtracting mean and dividing by standard deviation of the train set as a standard benchmark setting, following prior work [1,2].
>
> __Q5: How are lookback windows determined?__
>
> A5: Similar to prior work [1,2], we treat lookback windows as a hyperparameter. Section 3.2 and Appendix I describes and lists the hyperparameter values which we search through.
>
> __Q6: How much data was used for training in each case?__
>
> A6: Following prior work [1,2], we perform a 70/10/20 train-validation-test split. Section 3.2 describes our evaluation methodology which follows prior work. Listed below are the relevant information for each dataset.
>
> |   Dataset  |  ETTm2 |   ECL  | Exchange | Traffic | Weather | ILI |
> |:----------:|:------:|:------:|:--------:|:-------:|:-------:|:---:|
> | Dimensions |    7   |   321  |     8    |   862   |    21   |  7  |
> |  Timesteps | 57,600 | 26,304 |   7,588  |  17,544 |  52,696 | 966 |
>
> ---
> [1] Zhou, H., Zhang, S., Peng, J., Zhang, S., Li, J., Xiong, H., & Zhang, W. (2021, May). Informer: Beyond efficient transformer for long sequence time-series forecasting. In Proceedings of the AAAI Conference on Artificial Intelligence (Vol. 35, No. 12, pp. 11106-11115).
>
> [2] Challu, C., Olivares, K. G., Oreshkin, B. N., Garza, F., Mergenthaler, M., & Dubrawski, A. (2022). N-hits: Neural hierarchical interpolation for time series forecasting. arXiv preprint arXiv:2201.12886.
>
> [3] Wu, H., Xu, J., Wang, J., & Long, M. (2021). Autoformer: Decomposition transformers with auto-correlation for long-term series forecasting. Advances in Neural Information Processing Systems, 34, 22419-22430.
>
> [4] Zhou, T., Ma, Z., Wen, Q., Wang, X., Sun, L. &amp; Jin, R.. (2022). FEDformer: Frequency Enhanced Decomposed Transformer for Long-term Series Forecasting. <i>Proceedings of the 39th International Conference on Machine Learning</i>, in <i>Proceedings of Machine Learning Research</i> 162:27268-27286

---

> > ### Comment · Reviewer_3pqc · 2022-11-21
> > **Thank you for your response**
> >
> > Thank you for your response, which have addressed most of my earlier concerns -- raising my score accordingly.

---

> > > ### Author Response · Authors · 2022-11-25
> > > **Any further concerns**
> > >
> > > Dear Reviewer 3pqc,
> > >
> > > Thank you for taking the time for the review and acknowledgement. We are happy to have further discussions should you have any remaining concerns.
> > >
> > > Best regards,
> > >
> > > Authors

---

> ### Author Response · Authors · 2022-11-18
> **Response to Reviewer 3pqc (1/2)**
>
> Thank you Reviewer 3pqc for your comments and questions. Below is our response to your respective questions, and we hope that they fully address any concerns you may have had with our work.
>
> __Q1: Lack of hyperparameter optimization.__
>
> A1: We highlight that hyperparameters of DeepTime and benchmarks are ___not___ based on "suggested" settings, but rather, hyperparameter optimization.
>
> * Benchmark results are obtained from their respective papers [1,2,3,4] in which the authors have performed extensive hyperparameter tuning as described in their papers.
> * For DeepTime, we perform hyperparameter optimization as described in Section 3.2 and Appendix I. The presented hyperparameters in config files of our submitted code is in fact the results of hyperparameter optimization on the validation set. We present it as such for ease of reproducibility. Finally, the config file used for hyperparameter search is also presented in our submitted code, in ```experiments/configs/hp_search```.
>
> __Q2: Are improvements due to differences in features used? Specifically, do the random Fourier features introduce seasonal relationships which could also benefit other baselines (Transformer/Tabular models, and rolling linear regression on random Fourier features)?__
>
> A2: We highlight the following clarifications regarding random Fourier features:
>
> 1. Random Fourier features are not motivated by creating seasonal features as input. Rather, it has been noted in the Implicit Neural Representation literature that coordinate-based models suffer from "spectral bias". This means that when trying to reconstruct, for example, an image, the resulting reconstruction is typically blurry and smoothed (low frequency). Random Fourier features were proposed to solve this issue. Thus, this has become a common modification for coordinate-based models which we incorporate.
> 2. Inputs to other models already contain periodic features in the form of datetime features, including hour-of-day, day-of-week, etc., with a full list in Appendix N.1.
> 3. Random Fourier features are __not input dependent__. This means that regardless of the observations or datetime, the random Fourier features given to the model are always the same. Thus, in the context of Transformer based models, random Fourier featuers are better interpreted as a form of positional encodings, rather than features.
>
> That being said, we provide the requested ablations on several Transformers. To our understanding, a "rolling linear regression on random Fourier features" is equivalent to "+Local" in Table 3 of our ablation studies, which we include for your convenience.
>
> | Methods |     | DeepTime |       | FEDformer |       |  +RFF |       | Autoformer |       |     +RFF     |       | ETSformer |       |  +RFF |       | + Local |       |
> |:-------:|:---:|:--------:|:-----:|:---------:|:-----:|:-----:|:-----:|:----------:|:-----:|:------------:|:-----:|:---------:|:-----:|:-----:|:-----:|:-------:|:-----:|
> | Metrics |     |    MSE   |  MAE  |    MSE    |  MAE  |  MSE  |  MAE  |     MSE    |  MAE  |      MSE     |  MAE  |    MSE    |  MAE  |  MSE  |  MAE  |   MSE   |  MAE  |
> |  ETTm2  |  96 |   0.166  | 0.257 |   0.203   | 0.287 | 0.283 | 0.191 |    0.255   | 0.339 |     0.293    | 0.210 |   0.189   | 0.280 | 0.315 | 0.217 |  0.251  | 0.331 |
> |         | 192 |   0.225  | 0.302 |   0.269   | 0.328 | 0.329 | 0.263 |    0.281   | 0.340 |        0.324 | 0.265 |   0.253   | 0.319 | 0.348 | 0.273 |  0.322  | 0.371 |
> |         | 336 |   0.277  | 0.336 |   0.325   | 0.366 | 0.364 | 0.326 |    0.339   | 0.372 |     0.359    | 0.321 |   0.314   | 0.357 | 0.384 | 0.331 |  0.370  | 0.412 |
> |         | 720 |   0.383  | 0.409 |   0.421   | 0.415 | 0.425 | 0.428 |    0.422   | 0.419 |     0.419    | 0.418 |   0.414   | 0.413 | 0.441 | 0.437 |  0.443  | 0.449 |

---

### Official Review · Reviewer_i4vc · 2022-10-25

**Confidence:** 4
**Correctness:** 3
**Technical Novelty And Significance:** 2
**Empirical Novelty And Significance:** 2
**Recommendation:** 5

**Clarity, Quality, Novelty And Reproducibility:**

The writing was overall clear and relatively easy to follow, although the methodology and experiment descriptions lack many critical details (as detailed above) that make it difficult to assess or reproduce the method. I have questions on the novelty of the work due to its unclear and not-discussed relation with switching systems and more importantly, SNP.

Code is submitted and will be released for reproducibility.

**Strength And Weaknesses:**

Strengths:

1. Addressing non-stationary time-series by learning to adapt the time-series model is an important and interesting direction of research.
2. The use of INR for the deep time-index model is interesting.
3. The results demonstrated improvement over a good number of baselines used.


Weakness:

1. The presented work is heavily related to switching state-space models [1] and sequential neural processes [2]. In particular, it seems that it can be formulated as a special case of the SNP for learning y = f(t) where Bayesian meta-learning is learned to adapt f to observed frames from the lookback window for prediction in the forecasting horizon. Neither of these two works were discussed in the paper. Relations with these two works, and comparisons in terms of performance, should be provided.

2. By formulating the meta-learning with optimization-based approaches, the model needs to be optimized to the query set before being used for forecasting. To mitigate this, the paper opted to restrict the optimization to only the last layer of the INR. The effect of this restriction should be demonstrated, empirically at least. In an ideal setting (where resource or time is not a constraint) where the full INR can be optimized to the lookback window each time, how does the performance look like, and how does it compare with the assumption of restricting the optimization to the last layer? Note that, with SNP, this was avoided by formulating the meta-learning with feedforward based approaches. This again stresses the need to compare with SNP.

3. Details on meta-training is needed. It is not clear how the current tasks are defined -- by treating each pair of lookback window and forecasting horizon as a task, it means that the meta-training is looking at a large number of tasks and each task has only one set of context and query sets? Is the task boundary known assumed to be known ahead of time? If it is, it seems to be an unrealistic assumption. It it is not, this seems to be an unconventional setting of meta-learning -- Does the training follow the typical episodic training then? Further, how does the method apply if either the look-back window or the forecasting horizon falls within the transition of tasks boundaries?

4. Experimental details are missing. It is not clear how the baseline models are trained. Since most of them do not use any meta-learning formulation, are they trained on the meta-training set? In that case, the comparison may not be fair as the presented model -- at meta-test time, actually is optimized to the look-back window (while the baseline models are simply applied to the look-back window without optimization). A fair comparison would be to fine-tune the baseline models (after training on the meta-training set) to the same samples (look-back window) used at meta-test time. If the argument is that these models cannot be fine-tuned this way (since they are history-value based models), at least in ablation study, the "-RR" version need to be fine-tuned at meta-test time.

5. Since the look-back window represents context set in meta-learning, its size may have an important effect in "optimizing" the base model. The value of L in synthetic experiments seems to be large, and was not specified in real-data. Please add such details, and provide experimental evidence about the effect of the size of the context set L on meta-learning.

6. While the paper was heavily motivated for better learning non-stationary time series, the experimental evaluation is limited in demonstrating how or whether the proposed solutions achieved the stated goal. In synthetic experiments, there lack details on how many tasks were used to meta-train the model and how many tasks were used in meta-testing. Since tasks/segments are generated with random sampling of the parameters, it'd be good to get a sense of the distance between the meta-train and meta-testing tasks. Finally, it was stated that "A total of 400 time steps are sampled, with a lookback window length of 200 and forecast horizon of 200." In Appendix D, it was then stated that "each function/task consists of 400 evenly spaced points". So assuming each context-target set pairs are 400 time points, it was not clear how many such samples of length 400 were used. It was also not clear how does the forecasting work in such segment of 400. Does the model take 200 context samples and forecast for 200 context samples, and then it moves to the next window of 400? i.e., the task boundary is assumed to be known?

In real data experiments, it is understandable that the truth about the "non-stationary" nature of the data is not always available, but to the extent it's possible, it'd be desirable to say some level of analyses that link the model performance with the "non-stationary" nature of the underlying time series.



7. The methodology is presented in a general fashion for forecasting m-dimensional observations. It'd be good to understand to what extend the value of m could be, i.e., what types of observations can be modeled by the presented method. Are we looking at multivariate data with relatively lower number of dimensions, or are we looking at image series?






[1] Variational Learning for Switching State-Space Models, Zoubin Ghahramani and Geoffrey E. Hinton, Neural Computation 12, 831–864 (2000)
[2] Sequential Neural Processes, Gautam Singh, Jaesik Yoon, Youngsung Son, Sungjin Ahn, NeurIPS 2019



**Summary Of The Paper:**

This paper presents a meta-learning formulation for learning to adapt a deep time-index model to the look-back window. INR was the choice of time-index models, and meta-learning was formulated by using samples from look-back window as context set and forecasting horizon as query set. The goal of the meta-learning was mainly motivated for dealing with non-stationary time series. Experiments were conducted on synthetic as well as real datasets demonstrating performance improvement of forecasting over selected baselines.


**Summary Of The Review:**

This paper tackles an important problem of time-series forecasting (i..e, non-stationary series) with an interesting solution (meta-learning of time-index models). The novelty however is unclear with respect to some existing works especially sequential neural processes. The choice or benefit of meta-learning method and the simplification of optimizing only the last layer of the INR, in comparison to alternative meta-learning method such as feed-forward model based method used in SNP that can bypass such simplifications, is not clear. The writing lacks many critical details on methodology and experiments, such as how are tasks handled during meta-training, whether the task boundary needs to be known, and how to address lookback window or forecasting horizon that includes task boundaries. It is also not clear how large a L is needed as the context set, and how baseline models utilized these context data at meta-test time. Overall it is an interesting idea, but can be improved in clarification of novelty and many methodological/experimental details.

---

> ### Author Response · Authors · 2022-11-18
> **Response to Reviewer i4vc (4/4)**
>
> __Q6: Please provide more details on the synthetic experiments methodology.__
>
> A6: We highlight that the goal of the synthetic experiments, rather than to tackle the meta-learning problem, or to even use it in the standard time-series forecasting setting, the goal is to test the hypothesis, "Is DeepTime, trained on a family of functions following the same parametric form, able to perform extrapolation on unseen functions?"
>
> __Since tasks/segments are generated with random sampling of the parameters, it'd be good to get a sense of the distance between the meta-train and meta-testing tasks.__
>
> Meta-train and meta-test tasks were sampled independently from the same. We ensured that there is no overlap between the meta-train and meta-test set, by ensuring that there is no overlap in sampled parameters.
>
> __So assuming each context-target set pairs are 400 time points, it was not clear how many such samples of length 400 were used.__
>
> The meta-training set consists of 1000 samples.
>
> __It was also not clear how does the forecasting work in such segment of 400. Does the model take 200 context samples and forecast for 200 context samples, and then it moves to the next window of 400? i.e., the task boundary is assumed to be known?__
>
> The synthetic experiment is not temporal, the model takes 200 samples in the support set and 200 for the query set, and moves on to the next function.
>
> __Q7: In real data experiments, it is understandable that the truth about the "non-stationary" nature of the data is not always available, but to the extent it's possible, it'd be desirable to say some level of analyses that link the model performance with the "non-stationary" nature of the underlying time series.__
>
> A7: Thank you for raising this important point, please see our response to all reviewers, and also Appendix G.1 for an analysis of the non-stationarity of real-world datasets.
>
> __Q8: What types of observations can be modeled by the presented method. Are we looking at multivariate data with relatively lower number of dimensions, or are we looking at image series?__
>
> A8: We are interested in forecasting time-series, more specifically, metrics, numerical measurements of sensors, economic data, etc. Here are the descriptions of the real-world datasets used:
>
> |   Dataset  |  ETTm2 |   ECL  | Exchange | Traffic | Weather | ILI |
> |:----------:|:------:|:------:|:--------:|:-------:|:-------:|:---:|
> | Dimensions |    7   |   321  |     8    |   862   |    21   |  7  |
> |  Timesteps | 69,680 | 26,304 |   7,588  |  17,544 |  52,696 | 966 |
>
> * ETT - Electricity Transformer Temperature provides measurements from an electricity transformer such as load and oil temperature. We use the ETTm2 subset, consisting measurements at a 15 minutes frequency.
> * ECL - Electricity Consuming Load provides measurements of electricity consumption for 321 households from 2012 to 2014. The data was collected at the 15 mintue level, but is aggregated hourly.
> * Exchange - a collection of daily exchange rates with USD of eight countries (Australia, United Kingdom, Canada, Switzerland, China, Japan, New Zealand, and Singapore) from 1990 to 2016.
> * Traffic - from the California Department of Transportation providing the hourly road occupancy rates from 862 sensors in San Francisco Bay area freeways.
> * Weather - provides measurements of 21 meteorological indicators such as air temperature, humidity, etc., every 10 minutes for the year of 2020 from the Weather Station of the Max Planck Biogeochemistry Institute in Jena, Germany.
> * ILI - Influenza-like Illness measures the weekly ratio of patients seen with ILI and the total number of patients, obtained by the Centers for Disease Control and Prevention of the United States between 2002 and 2021.
> ___
>
> [1] Dahlhaus, R. (2012). Locally stationary processes. In Handbook of statistics (Vol. 30, pp. 351-413). Elsevier.
>
> [2] Zhou, H., Zhang, S., Peng, J., Zhang, S., Li, J., Xiong, H., & Zhang, W. (2021, May). Informer: Beyond efficient transformer for long sequence time-series forecasting. In Proceedings of the AAAI Conference on Artificial Intelligence (Vol. 35, No. 12, pp. 11106-11115).
>
> [3] Challu, C., Olivares, K. G., Oreshkin, B. N., Garza, F., Mergenthaler, M., & Dubrawski, A. (2022). N-hits: Neural hierarchical interpolation for time series forecasting. arXiv preprint arXiv:2201.12886.
>
> [4] Wu, H., Xu, J., Wang, J., & Long, M. (2021). Autoformer: Decomposition transformers with auto-correlation for long-term series forecasting. Advances in Neural Information Processing Systems, 34, 22419-22430.
>
> [5] Zhou, T., Ma, Z., Wen, Q., Wang, X., Sun, L. &amp; Jin, R.. (2022). FEDformer: Frequency Enhanced Decomposed Transformer for Long-term Series Forecasting. <i>Proceedings of the 39th International Conference on Machine Learning</i>, in <i>Proceedings of Machine Learning Research</i> 162:27268-27286

---

> > ### Comment · Reviewer_i4vc · 2022-11-30
> > **Thanks for the response**
> >
> > I thank the authors for a substantial amount of work for the rebuttal, especially the added baselines and ablation studies.
> >
> > Many of these additions and revisions however needed to be included in the revised manuscript. Furthermore, several of my concerns remains. I will thus maintain my original rating.
> >
> > 1. I’m still not fully convinced by the argument of not needing task boundaries and the associated fundamental assumptions: that the authors assumed that y(t) samples in each pair of lookback-horizon window is generated by one function/task (while different pairs are described by different tasks). As a result, the meta-training consists a number of tasks as large as the training pairs. More importantly, the same y(t) sample is assumed to be described by many different functions/tasks. The authors argue that this is a novel extension of meta-learning, yet to me some theoretical backing is needed to show that the meta-model can learn to pull knowledge across such a large number of tasks (practically unlimited as the number of tasks increases with the training pairs) with overlapping samples (and a single or limited split of context./target set per task). More importantly, how would the model behave as the test tasks become increasingly different from the training tasks (see 4 below).
> >
> >
> > At the same time, while it is true that the model does not need known task boundaries, there is the assumption that all samples within a pair of window are from the same task: if we look at a pair of windows where the samples of y(t)’ maybe generated by two different functions, the assumption is actually violated.
> >
> > Thus overall, the fundamental assumption or advantage of treating each look back-horizon pair as a task is not theoretically convincing to me.
> >
> >
> > 2. The authors argued that switching dynamics models requires predefined regimes. I’m not entirely convinced by this argument. Many models of switching dynamics also continuously infer the time-varying switching variables from the current window of observations. Perhaps the authors can clarify.
> >
> > 3. In my earlier comment, I meant to relate the proposed work to neural process (NP) instead of SNP. If we consider a neural process y(t), a context embedding can be derived from lookback window samples of y and t to learn the context embedding, which can be used to condition the regression function y as a function of (t) to be applied on the horizon window. While not used in the NP’s original work, the same assumption used here can still be adopted: that each pair of such window is one task on its own. Perhaps the authors can clarify why the NP cannot directly extended to learn this time index model y(t).
> >
> > 4. In meta-learning, commonly one would like to see a difference in distribution between meta-training and meta-test tasks. Most of the time, some “unseen” tasks will be held out on purpose for meta-testing. While the authors showed statistics for the distributions of test samples regarding the non-stationary nature of the data, it is still not clear to me how far are the distribution between meta-training and -test samples. In synthetic data, the authors clarified that training and test data are sampled from the same distribution, which means there are no controlled distributional differences. In real data, this was also not clear. For a meta-learning formulation, it’d be important to show that the model actually learned how to generalize to tasks far away from training tasks (rather than memorized how to be adjusted to a task).
> >
> > 5. The size of the lookback window (context set) appears large for meta-learning purpose. How small can the context set go?
> >
> > 6. As a minor note, there has been some recent work of meta forecasting using historical-valued models. E.g., Meta-learning dynamics forecasting using task inference (https://arxiv.org/abs/2102.10271), where the task is inferred from each lookback window. This may be a strong baseline for the proposed work.

---

> > > ### Author Response · Authors · 2022-12-01
> > > **Further Response (3/3)**
> > >
> > > __Q5: The size of the lookback window (context set) appears large for meta-learning purpose. How small can the context set go?__
> > >
> > > The size of the lookback window can technically be any size, please see the below experiments to understand how performance scales as size decreases. We note further deterioration as lookback window length decreases (worse than SOTA baselines), but these are smaller than usual values used for forecasting.
> > >
> > > | Horizon |   96  |       |  192  |       |  336  |       |  720  |       |
> > > |:-------:|:-----:|:-----:|:-----:|:-----:|:-----:|:-----:|:-----:|:-----:|
> > > |   $\mu$  |  MSE  |  MAE  |  MSE  |  MAE  |  MSE  |  MAE  |  MSE  |  MAE  |
> > > |   0.1   | 0.270 | 0.350 | 0.313 | 0.386 | 0.381 | 0.426 | 0.524 | 0.508 |
> > > |   0.3   | 0.220 | 0.318 | 0.279 | 0.356 | 0.365 | 0.413 | 0.440 | 0.452 |
> > > |   0.5   | 0.202 | 0.297 | 0.269 | 0.345 | 0.333 | 0.389 | 0.398 | 0.422 |
> > > |   0.7   | 0.193 | 0.287 | 0.264 | 0.341 | 0.300 | 0.360 | 0.404 | 0.427 |
> > > |   0.9   | 0.193 | 0.288 | 0.253 | 0.330 | 0.310 | 0.369 | 0.398 | 0.425 |
> > >
> > > __Q6: As a minor note, there has been some recent work of meta forecasting using historical-valued models. E.g., Meta-learning dynamics forecasting using task inference (https://arxiv.org/abs/2102.10271), where the task is inferred from each lookback window. This may be a strong baseline for the proposed work.__
> > >
> > > A6: Thank you for highlighting this work. Upon reading this paper, we understand that the problem setting and proposed solution is quite different which would make it inappropriate for the time-series forecasting task:
> > >
> > > (1) Specifically proposed for dynamics forecasting. Their major contribution involves architecture design and specialized modules for processing 4d inputs/outputs, and not applicable for time-series data.
> > > (2) Their meta-learning formulation requires explicit task partitions and knowledge of task parameters (for weak supervision). Removing these components and applying this method directly on time-series forecasting would result in just a simple supervised learning model.
> > >
> > > Finally, based on the questions posed by the reviewer, we surmise that they may have some qualms regarding the strength of existing baselines used in our paper. We reiterate that the existing baselines used are all existing SOTA methods for the long sequence time-series forecasting setting, published in top conferences (NeurIPS'22, ICML'22, etc..) [1,2,3,4,5], with [5] being our most relevant competitor which aims to solve the problem of non-stationarity in long sequence time-series forecasting.
> > >
> > > ---
> > > [1] Zhou, H., Zhang, S., Peng, J., Zhang, S., Li, J., Xiong, H., & Zhang, W. (2021, May). Informer: Beyond efficient transformer for long sequence time-series forecasting. In Proceedings of the AAAI Conference on Artificial Intelligence (Vol. 35, No. 12, pp. 11106-11115).
> > >
> > > [2] Challu, C., Olivares, K. G., Oreshkin, B. N., Garza, F., Mergenthaler, M., & Dubrawski, A. (2022). N-hits: Neural hierarchical interpolation for time series forecasting. arXiv preprint arXiv:2201.12886.
> > >
> > > [3] Wu, H., Xu, J., Wang, J., & Long, M. (2021). Autoformer: Decomposition transformers with auto-correlation for long-term series forecasting. Advances in Neural Information Processing Systems, 34, 22419-22430.
> > >
> > > [4] Zhou, T., Ma, Z., Wen, Q., Wang, X., Sun, L. &amp; Jin, R.. (2022). FEDformer: Frequency Enhanced Decomposed Transformer for Long-term Series Forecasting. <i>Proceedings of the 39th International Conference on Machine Learning</i>, in <i>Proceedings of Machine Learning Research</i> 162:27268-27286
> > >
> > > [5] Yong Liu, Haixu Wu, Jianmin Wang, and Mingsheng Long. Non-stationary transformers: Exploring the stationarity in time series forecasting. In Alice H. Oh, Agarwal, Danielle Belgrave, and Kyunghyun Cho (eds.), Advances in Neural Information Processing Systems, 2022. URL https://openreview.net/forum?id=ucNDIDRNjjv.
> > >
> > > [6] Finn, C., & Levine, S. (2017). Meta-learning and universality: Deep representations and gradient descent can approximate any learning algorithm. arXiv preprint arXiv:1710.11622.
> > >
> > > [7] Dahlhaus, R. (2012). Locally stationary processes. In Handbook of statistics (Vol. 30, pp. 351-413). Elsevier.

---

> > > ### Author Response · Authors · 2022-12-01
> > > **Further Response (2/3)**
> > >
> > > __Q2: Authors argued that switching dynamics models requires predefined regimes, many models of switching dynamics also continuously infer the time-varying switching variables from the current window of observations. Perhaps the authors can clarify.__
> > >
> > > A2: Apologies for this vague description, we meant to highlight that switching state-space models follow the assumption of discrete states, of which the total possible states is some predefined hyperparameter $K$. Such an assumption does not generalize well to unseen states, as well as when the data-generating distribution is gradually changing, which is better supported by continuous state transitions.
> > >
> > > __Q3: Why are Neural Processes not able to directly extend to learn a time-index model y(t)?__
> > >
> > > A3: We highlight again that the task at hand is specifically non-stationary time-series forecasting.
> > > 1. Neural Processes (NPs) are in fact more analogous to historical-value models, than time-index models. Similar to SNPs, they rely on a "blackbox meta-learning" methodology, which aggregate observations into a representation via a neural network. This means that NPs take as input, into a neural network, the observations from the lookback window. As such, they suffer the same drawbacks as historical-value models as listed Section 1, namely they would suffer from covariate shift.
> > > 2. Building upon the point about "blackbox meta-learning", NPs specifically follow a latent variable model methodology, rather than solve the bi-level optimization problem as described in Section 2.1, Equations (1,2). Existing studies [6] have shown that optimization/gradient-based meta-learning methods generalize better than blackbox-based methods, reason being that explicit optimization steps at meta-test time intuitively is stronger than a single forward pass.
> > > 3. Finally, we highlight that our contributions not only lie in this meta-learning formulation, but we also show how to effectively use time-index models for forecasting. This includes how we use Implicit Neural Representations, and our novel concatenated Fourier features module. Overall, we believe it is plausible that a method using the general framework of NPs is able to achieve competitive forecasting performance. However, in it's current form as described the current literature, they are unable to do so, and it would take a non-trivial amount of effort to introduce adaptations (architecture, training method, etc.) to make it viable for time-series forecasting.
> > >
> > > __Q4: In meta-learning, commonly one would like to see a difference in distribution between meta-training and meta-test tasks. Most of the time, some “unseen” tasks will be held out on purpose for meta-testing. While the authors showed statistics for the distributions of test samples regarding the non-stationary nature of the data, it is still not clear to me how far are the distribution between meta-training and -test samples. In synthetic data, the authors clarified that training and test data are sampled from the same distribution, which means there are no controlled distributional differences. In real data, this was also not clear. For a meta-learning formulation, it’d be important to show that the model actually learned how to generalize to tasks far away from training tasks (rather than memorized how to be adjusted to a task).__
> > >
> > > A4: We highlight that we are __not__ attempting to solve a meta-learning problem. Instead, we are using meta-learning as a tool, to solve the problem of non-stationarity in time-series forecasting. We use a meta-learning formulation to enable time-index models to perform forecasting, and shown that this unique combination leads to a strong solution to non-stationary forecasting, beating the recently proposed [5].
> > > Thus, we once again urge the reviewer to evaluate our work in the lens of long sequence time-series forecasting, where we have provided strong empirical evidence on the strength and efficiency of our new proposed method. Our motivation and end goal is __time-series forecasting__.
> > >
> > > Furthermore, we note that our performed experiments are highly similar to that in the Neural Process and Conditional Neural Process papers, in which there was no controlled distributional differences. Both train and test sets were generated from Gaussian Process with the same hyperparameters. We refer to the source code: https://github.com/deepmind/neural-processes/blob/master/conditional_neural_process.ipynb.

---

> > > > ### Comment · Reviewer_i4vc · 2022-12-01
> > > > **Thanks for the response. I will keep my rating.**
> > > >
> > > > Thanks for your response, and I truly appreciate the effort and patience taken in these discussion.
> > > > The clarification that tau_t changes for y(tau_t) in each window/task is very helpful and clears a main concern of mine.
> > > >
> > > > My two other primary concerns however remain.
> > > >
> > > > 1. As mentioned in my last question (and indicated by the authors in the answers), we can think of the regression function y(t) as a NP where the (y(tau_t), tau_t) pairs in the lookback window serve as the context data, to condition a forecaster to forecasting the horizon window. Note that each sample would be the (y(tau_t), tau_t) pair, not all historical values in the window. It seems to me that the original NP can be directly extended for this purpose (and NP is originally designed for this type of regression problems), and the authors' response did not provide a concrete justification for why this is not the case. Model-based and optimization-based meta-learning are both popular meta-learning approaches with their pros and cons, and I don't think general arguments about one being better provide convincing justification to avoid comparisons in a specific problem context.
> > > >
> > > > I understand that the paper's main contribution is to improve time-series forecasting leveraging meta-learning as a novel solution, but the significance of the contribution/novelty needs to be justified by showing that it cannot be achieved by a direct use or extension of existing meta-learning frameworks such NP.
> > > >
> > > > 2. Non-stationary time series is a key motivation of the paper. While general statistics are provided for the datasets, my question about the evidence of the co-variate and distributional shift between training and test data is not resolved.

---

> > > > > ### Author Response · Authors · 2022-12-05
> > > > > **Response to remaining concerns (2/2)**
> > > > >
> > > > > __Q2: While general statistics are provided for the datasets, my question about the evidence of the co-variate and distributional shift between training and test data is not resolved.__
> > > > >
> > > > > A2: We highlight that presented analyses of the real-world datsets (response to all reviewers + Appendix G.1) are __not simply general statistics of the dataset__. We present results of statistical tests which provide __evidence of covariate and conditional distribution shift between the train and test sets__.
> > > > >
> > > > > In particular, the Augmented Dickey-Fuller (ADF) test, tests the null hypothesis that a unit root is present in a time series sample. A stochastic process with a unit root is non-stationary (equivalent to our definition of covariate shift).
> > > > > We follow up with visual analysis in Figure 6, which shows the marginal histogram plots __between train and test sets__, which also help to verify the (1st-order) covariate shift.
> > > > >
> > > > > Next, we perform a Chow test between the train and test regions. The Chow tests helps test whether the true coefficient in two linear regressions on the __train and test regions__ are equal, signifying a conditional distribution shift, as we use an autoregressive formulation for the linear regression test.
> > > > >
> > > > > Finally, we reiterate that existing work [1] (NeurIPS'22) has used the same real-world for empirical verification of their work on non-stationary time-series forecasting.
> > > > >
> > > > > ---
> > > > > [1] Yong Liu, Haixu Wu, Jianmin Wang, and Mingsheng Long. Non-stationary transformers: Exploring the stationarity in time series forecasting. In Alice H. Oh, Agarwal, Danielle Belgrave, and Kyunghyun Cho (eds.), Advances in Neural Information Processing Systems, 2022. URL https://openreview.net/forum?id=ucNDIDRNjjv.

---

> > > > > > ### Comment · Reviewer_i4vc · 2022-12-12
> > > > > > **Thanks**
> > > > > >
> > > > > > Thanks to the authors for the extensive discussion and effort during the rebuttal period. I think this work will be interesting to the community, but the manuscript in its current form requires too substantial a revision to be accepted without further peer-review -- even just to include the new results added since the last revision. I'd thus maintain my current rating.
> > > > > >
> > > > > > I encourage the authors to address these two points carefully before the manuscript is received for the next round of peer review -- 1) to conduct a solid comparison study with NP formulation of the problem, where the architecture of the neural network actually matches the presented model. This will provide the best evidence regarding the difference between the MAML vs. model-based meta-learning formulation of the presented work; 2) to have some experiments where the non-stationary of the data are clearly designed and one can clearly appreciate the change of the underlying generation function at training vs. test time; carefully designed synthetic data could be a good candidate for this.

---

> > > > > ### Author Response · Authors · 2022-12-05
> > > > > **Response to remaining concerns (1/2)**
> > > > >
> > > > > Thank you Reviewer i4vc for taking the time to follow up, we really appreciate your detailed feedback and suggestions, it has helped us to continually improve our work. We are glad that our replies have managed to clarify some of your concerns. We hope that the following clears up any remaining concerns and doubts, and we are always happy for further discussion.
> > > > >
> > > > > __Q1: The original NP can be directly extended for time-series forecasting (and NP is originally designed for this type of regression problems). The significance of the contribution/novelty needs to be justified by showing that it cannot be achieved by a direct use or extension of existing meta-learning frameworks such NP.__
> > > > >
> > > > > A1: Thank you for raising up this important related method. Our work was originally motivated from the perspective of time-index models and inspired by implicit neural representations, thus we left out this crucial comparison. We ran further experiments implementing both Neural Processes (deterministic + latent) and Conditional Neural Processes, and the results are as follows. __We fully intend to include these in the revised manuscript.__
> > > > >
> > > > > |  CNP  |       |   NP   |       |
> > > > > |:-----:|:-----:|:------:|:-----:|
> > > > > |  MSE  |  MAE  |   MSE  |  MAE  |
> > > > > | 1.335 | 0.830 |  0.824 | 0.742 |
> > > > > | 1.274 | 0.815 |  1.642 | 0.985 |
> > > > > | 2.888 | 1.352 |  0.953 | 0.766 |
> > > > > | 4.012 | 1.708 |  3.686 | 1.640 |
> > > > > | 0.928 | 0.793 |  0.926 | 0.792 |
> > > > > | 0.945 | 0.792 |  0.967 | 0.797 |
> > > > > | 0.952 | 0.794 |  0.980 | 0.800 |
> > > > > | 0.959 | 0.811 |  0.962 | 0.807 |
> > > > > | 0.947 | 0.829 |  1.120 | 0.876 |
> > > > > | 3.350 | 1.328 |  1.385 | 0.929 |
> > > > > | 5.161 | 1.818 | 12.719 | 2.907 |
> > > > > | 5.879 | 2.061 |  1.152 | 0.885 |
> > > > > | 1.433 | 0.773 |  1.465 | 0.805 |
> > > > > | 1.461 | 0.796 |  1.467 | 0.801 |
> > > > > | 1.477 | 0.800 |  1.489 | 0.805 |
> > > > > | 1.497 | 0.808 |  1.527 | 0.814 |
> > > > > | 0.329 | 0.374 |  0.288 | 0.352 |
> > > > > | 0.454 | 0.463 |  0.712 | 0.541 |
> > > > > | 0.455 | 0.465 |  0.468 | 0.459 |
> > > > > | 0.451 | 0.481 |  0.456 | 0.470 |
> > > > > | 4.863 | 1.536 |  5.192 | 1.588 |
> > > > > | 6.589 | 1.790 |  6.169 | 1.689 |
> > > > > | 4.862 | 1.541 |  5.062 | 1.573 |
> > > > > | 6.921 | 1.840 |  5.681 | 1.642 |
> > > > >
> > > > > We observe that NPs and CNPs have difficulty in performing the forecasting task, something that they have not been proposed for. We surmise that the disappointing performance stems from 2 reasons: (1) encoder suffers from covariate shift, as it takes as input, time-series values, which shift across train set, and from train to test set. (2) their architecture follows a standard MLP -- there needs to be more advanced modifications to perform better for the forecasting task.
> > > > >
> > > > > We hope that this empirical backing convinces the reviewer on the novelty and strength of our work, that the similar scheme of Neural Processes as proposed originally are insufficient for the forecasting task.
> > > > >
> > > > > Furthermore, we will be __adding the following in related work of the revised manuscript, to elaborate more about NPs__:
> > > > >
> > > > > _Neural Processes (NPs) and Conditional Neural Processes (CNPs) for 1d regression follow a similar scheme to time-index models, mapping a coordinate to the value at that coordinate. Different from DeepTime, NPs and CNPs are optimized via a model-based meta-learning approach, rather than an optimization-based meta-learning method. This means that they are comprised of an encoder which takes as input the coordinate and time-series value pairs, thus, they are in fact a historical-value model.
> > > > > Similar to Gaussian processes, NPs and CNPs are also capable of probabilistic forecasts, and typically model a multivariate Gaussian distribution. NPs and CNPs differ in that NPs are latent variable models, explicitly modelling a latent variable representing the context set, whereas CNPs treat the context set representation as purely deterministic._
> > > > >
> > > > > Finally, while the current results for directly extending NPs and CNPs are not competitive, we believe they are still a viable approach which may bring about certain benefits over optimization-based methods (e.g. single forward pass rather than optimization steps). Yet, non-trivial effort and novelty is required to adapt these methods to enable competitive results. We think the following directions would be interesting future work:
> > > > > 1. Further improvements to architecture of NPs and CNPs:
> > > > >     1. Further studies into normalization techniques to avoid covariate shift problem of the encoders. Time-series data (different from CV and NLP) have the added problem of the data scaling. (Language have discrete tokens, image pixels are bounded depending on encoding scheme, but time-series data can take on reals, integers, etc.)
> > > > >     2. More powerful approaches to integrating context representations into the decoder at an model architecture level, e.g. will a hypernetwork style architecture better integrate context representations to the decoder rather than concatenation?
> > > > > 2. Studies comparing data scaling capabilities between optimization-based vs blackbox-based meta-learning approaches.

---

> > > > > ### Author Response · Authors · 2022-12-07
> > > > > **Gentle reminder**
> > > > >
> > > > > Dear Reviewer i4vc,
> > > > >
> > > > > Thank you for your review and feedback on our response to your comments. We appreciate the time and effort you put into providing detailed and constructive criticism, and we are committed to addressing your concerns in a thorough and thoughtful manner.
> > > > >
> > > > > As the deadline for the discussion period is approaching, we wanted to check in with you to see if you have any further updates or comments on our response. If there are any remaining concerns or questions that you would like us to address, we would be happy to do so. We apologize for reminding you so early, but we want to make sure that we have enough time to fully address any remaining issues before the deadline.
> > > > >
> > > > > Thank you again for your review, and we look forward to hearing from you.

---

> > > > > ### Author Response · Authors · 2022-12-11
> > > > > **Gentle reminder**
> > > > >
> > > > > Dear Reviewer i4vc,
> > > > >
> > > > > Hope everything is well. Thank you for all the time and effort spent in reviewing thus far. We would like so send a gentle reminder since the discussion period is coming to an end very soon. We have addressed your final 2 concerns:
> > > > >
> > > > > 1. Including neural processes as a baseline comparison (qualitative and quantitative).
> > > > > 2. Evidence of covariate and conditional distribution shift in train and test regions.
> > > > >
> > > > > Hope that you are able to review these changes, thank you very much!

---

> > > ### Author Response · Authors · 2022-12-01
> > > **Further Response (1/3)**
> > >
> > > Thank you Reviewer i4vc for taking the time and effort for the review and response. All additional ablations and revisions __have already been included in the revised manuscript__, and we fully intend to include the 2 additional baselines (which were only completed after the manuscript revision deadline).
> > >
> > > We highlight that our work is targetting the ___time-series forecasting setting___, and more specifically, the line of work on Long Sequence Time-series Forecasting. The most relevant comparisons are the recent SOTA baselines that we have considered [1,2,3,4,5], and we highlight that the problem setting we work in is __identical__ to these works. We are not targetting a new meta-learning problem. Instead, we use meta-learning as a solution to solve the problem of non-stationarity in time-series forecasting (also highlighted in [5]), to achieve a strong and efficient time-series forecasting model. Thus, we urge the reviewer to look at our work in the light of long sequence time-series forecasting, rather than a pure meta-learning paper.
> > >
> > > __Q1a: More importantly, the same y(t) sample is assumed to be described by many different functions/tasks. The authors ue that this is a novel extension of meta-learning, yet to me some theoretical backing is needed to show that the meta-model can learn to pull knowledge across such a large number of tasks with overlapping samples.__
> > >
> > > A1a: Consider the 1d regression toy experiment commonly used in Neural Process literature. Furthermore, the line $y = a * sin(b * 2\pi * x)$, where $a, b$ are randomly sampled to obtain new tasks. Suppose the point $x = 0$ appears in multiple context sets. Then in this case $(x = 0, y = 0)$ holds no matter the task, i.e. no matter what values of $a, b$ are sampled. This is one such example where samples can fall into different tasks.
> > >
> > > __Importantly__, note that while $y(t)$ may fall into multple lookback windows, it's corresponding value of $x(t)$, or in our notation $\tau_t$, will be different. Thus, we note that $(x,y)$ pair does __not__ appear in multiple contexts.
> > >
> > > __Q1b: How can the model learn from such a large number of tasks?__
> > >
> > > A1b: We highlight that we are tackling the time-series forecasting problem, and in this setting, windows of time-series are highly related, and we can see them as belonging to the same task distribution. In fact, with access to more tasks, meta-learning becomes easier.
> > >
> > > __Q1c: More importantly, how would the model behave as the test tasks become increasingly different from the training tasks?__
> > >
> > > A1c: As test tasks becomes increasingly different from training tasks, we would expect that the model would deteriorate, and we believe that this is a very reasonable outcome.
> > >
> > > __Q1d: At the same time, while it is true that the model does not need known task boundaries, there is the assumption that all samples within a pair of window are from the same task: if we look at a pair of windows where the samples of y(t)’ maybe generated by two different functions, the assumption is actually violated.__
> > >
> > > A1d: Indeed, we use the locally stationary assumption [7] which would be violated if samples from the same window were from a different underlying distribution. However, in all machine learning methods, assumptions are violated. Regular forecasting methods which do not account for non-stationarity implicitly assume that the whole time-series dataset comes from the same data-generating distribution. This is a much larger violation of assumption.
> > >
> > > The main question is whether under violation of these assumptions, does the model still perform well? And again, we stress that our proposed method is backed by strong empirical results, which shows that regardless of assumptions being violated, it is still able to perform well for it's intended task.

---

> ### Author Response · Authors · 2022-11-18
> **Response to Reviewer i4vc (3/4)**
>
> __Q4: Please provide further experimental details, in particular, how are baseline models trained? Do they use a meta-learning formulation?__
>
> A4: Baseline results are lifted from their respective papers, meaning that they are trained as per their authors describe, via standard gradient descent methods.
>
> We respectfully disagree that the comment that "the comparison may not be fair as the presented model ... optimized to the lookback window, while the baseline models are simply applied ... without optimization", with our reasons as follows:
> 1. At train/test time, the information given to DeepTime and all baselines are the same, a lookback window of length $L$, and optionally, the datetime information.
> 2. The reason is that our model is able to do optimization steps because it is a __time-index model__, whereas baselines are __historical-value models__. Time-index models are able to construct $L$ samples from a lookback window of length $L$, whereas a historical-value model only has 1 sample from the same lookback window.
> 3. As the reviewer points out, historical-value models are not able to be trained via meta-learning formulation/fine-tuned at test time because it leads to infeasible computation cost. Observe from our efficiency analysis, that baselines are already much more computationally expensive than DeepTime.
>
> The reviewer mentions a version of the ablation study which includes a fine-tuning at meta-test time. "+Local" performs training on each lookback window at test time (from scratch), and we further include "+Finetune" by first training on the train set, then perform finetuning at meta-test time. We note similar results to the -RR setting with some marginal improvements.
>
> | Methods |     |  DeepTime |           | + Finetune |       | + Finetune |       |
> |:-------:|:---:|:---------:|:---------:|:----------:|:-----:|:----------:|:-----:|
> |         |     |           |           |            |       | + Datetime |       |
> | Metrics |     |    MSE    |    MAE    |     MSE    |  MAE  |     MSE    |  MAE  |
> |  ETTm2  |  96 | **0.166** | **0.257** |    3.028   | 1.328 |    3.242   | 1.365 |
> |         | 192 | **0.225** | **0.302** |    3.043   | 1.341 |    3.385   | 1.391 |
> |         | 336 | **0.277** | **0.336** |    2.950   | 1.331 |    3.367   | 1.387 |
> |         | 720 | **0.383** | **0.409** |    2.721   | 1.253 |    3.476   | 1.407 |
>
>
> __Q5: Please further elaborate on how the lookback length $L$ is chosen, and the effect of it's size.__
>
> A5: In the real-world data experiments, as per the established convention, the size of the lookback window, $L$, is treated as a hyperparameter. As specified in Section 3.2 and Appendix I, we search through values of $L$ as a multiple of the horizon length $H$, by setting $L = \mu * H$, for $\mu = \\{1, 3, 5, 7, 9\\}$.
>
> Below is a sensitivity analysis on the value of $\mu$ across various horizons on the ETTm2 dataset. We observe that for shorter horizons, a larger value of $\mu$ is preferred while for longer horizons, a smaller value of $\mu$ is preferred, indicating an optimal lookback length which is similar regardless of horizon length.
>
> | Horizon |     96    |           |    192    |           |    336    |           |    720    |           |
> |:-------:|:---------:|:---------:|:---------:|:---------:|:---------:|:---------:|:---------:|:---------:|
> |  $\mu$  |    MSE    |    MAE    |    MSE    |    MAE    |    MSE    |    MAE    |    MSE    |    MAE    |
> |    1    |   0.192   |   0.287   |   0.255   |   0.332   |   0.294   |   0.354   | **0.383** | **0.409** |
> |    3    |   0.172   |   0.264   |   0.228   |   0.304   |   0.277   | **0.336** |   0.371   |   0.403   |
> |    5    |   0.168   |   0.259   |   0.225   |   0.302   | **0.275** |   0.337   |   0.389   |   0.420   |
> |    7    |   0.166   |   0.257   | **0.223** | **0.300** |   0.279   |   0.343   |   0.440   |   0.451   |
> |    9    | **0.165** | **0.258** | **0.223** |   0.301   |   0.285   |   0.350   |   0.409   |   0.434   |

---

> ### Author Response · Authors · 2022-11-18
> **Response to Reviewer i4vc (2/4)**
>
> __Q2: Please provide comparisons to the ideal setting where the full INR can be optimized.__
>
> A2: We provide ablations on the full MAML setting below. Inner loop optimization is performed using the Adam optimizer, and is tuned over lookback length multiplier values of $\{1,3,5,7,9\}$, and inner loop iterations of $\{1,5,10\}$. While we may expect a full MAML to always outperform a restriction to only optimize the last layer, in reality, there are many complications in training a full MAML based solution which may lead to suboptimal results. Firstly, MAML is sensitive to neural network architectures, and are difficult to optimize, often leading to instability during training. They also require arduous hyperparameter searches to stabilize training and achieve high generalization. Last layer optimization provides a useful prior which enables stable optimization and easily achieves high generalization without arduous hyperparameter search.
>
> | Methods |     | DeepTime |       | Full MAML |       |
> |:-------:|:---:|:--------:|:-----:|:---------:|:-----:|
> |         |     |          |       |           |       |
> | Metrics |     |    MSE   |  MAE  |    MSE    |  MAE  |
> |  ETTm2  |  96 |   0.166  | 0.257 |   0.235   | 0.326 |
> |         | 192 |   0.225  | 0.302 |   0.295   | 0.361 |
> |         | 336 |   0.277  | 0.336 |   0.348   | 0.392 |
> |         | 720 |   0.383  | 0.409 |   0.491   | 0.484 |
>
> __Q3: Please provide further details on meta-training.__
>
> A3: In the time-series forecasting setting, we have a time-series dataset, $y_1, \ldots y_T$. This can be split into overlapping lookback-horizon pairs, $(y_{t-L:t}, y_{t:t+H})$, where $y_{t:t-L} = (y_{t-L}, \ldots, y_{t-1})$, and $y_{t:t+H} = (y_t, \ldots y_{t+H-1})$, for $t = L+1, \ldots, T-H+1$. We treat each lookback-horizon pair as a task.
> 1. During meta-training, from each task, we obtain a single support/query set pair. However, this can easily be generalized into a formulation where we subsample the lookback window and forecast horizon to obtain more support/query sets.
> 2. Task boundaries are not assumed to be known. As observed in the above formulation, a single time step can be part of multiple tasks. Since lookback window is treated as a hyperparameter (consistent with existing work), task boundary is essentially cross-validated.
> 3. Rather than episodic training, we are able to batch multiple tasks for meta-training.
>
> We acknowledge that this is an unconventional (novel) meta-learning formulation, and is one of our contributions in formulating deep time-index meta-learning to tackle the standard problem setting of time-series forecasting. We have updated our manuscript with these information, which can be found in Section 2, 2.1, and equations (1,2).

---

> ### Author Response · Authors · 2022-11-18
> **Response to Reviewer i4vc (1/4)**
>
> Thank you Reviewer i4vc for taking the time to read our paper and giving detailed feedback and questions. Please see below for our response to your specific questions, we hope they fully address any concerns and queries you have, and remain committed to address any further issues.
>
> __Q1: Please provide comparisons to (1) switching state-space models, and (2) sequential neural processes.__
>
> A1: Thank you for raising these work up, we have now updated our manuscript to include the mentioned work in our extended related works section. Please see the following for some discussion comparing DeepTime with the two mentioned methods:
>
> __Switching State-Space Models__ Switching state-space models generalize and combine hidden Markov models and state-space models, where the dynamics in each regime is typically represented by a linear model (linear dynamic system) and switches between regimes, controlled by hidden transition probabilities of a Markov process. While they can be applied to non-stationary time-series, they make additional assumptions, such as a predefined of segments or regimes. In time-series forecasting, we do not have such knowledge. Furthermore, the data generating distribution could be gradually changing, or frequently changing. DeepTime on the other hand, does not much such assumptions, but continuously infers the local parameters given the current lookback window.
>
> __Sequential Neural Process__ SNPs incorporates a temporal state-transition model of stochastic processes, extending Neural Process framework to dynamic stochastic processes. Importantly, SNPs use a "black-box meta-learning" approach, while DeepTime uses an "optimization-based meta-learning" approach. Further differences include that the standard SNP setting requires knowledge of task boundaries, and multiple support/query sets per task. This makes it a non-trivial task to adapt SNP for time-series forecasting, since the original setting it was proposed for is significantly different.
> Thus, we respectfully argue that the characterization of DeepTime being a special case of SNP is rather tenuous, since the fundamental approach to meta-learning is different, as with the assumptions behind the methods.
>
> Finally we highlight that our work aims to target the Long Sequence Time-series Forecasting problem setting [2,3,4,5], which is characterized by forecasting over long horizons. Models such as RNNs on which switching SSMs and SNPs leverage, face severe drawbacks (computational efficiency and long range dependencies) when tackling this setting.
>
> We are currently running experiments for these two requested baselines, and request for your kind patience as we work to produce these results.

---

> > ### Author Response · Authors · 2022-11-19
> > **Update: Preliminary results for baselines**
> >
> > Please see below for some preliminary results on the ETTm2 dataset. We have implemented a recent switching state space model, SNLDS [2], as well as SNP. For the SNP we generated "tasks" by rolling through the time-series dataset. Each task consist of a sequence of 9 stochastic process (heuristically chosen hyperparam), and each stochastic process is made up of a lookback window and forecast horizon. Thus, each "task" the model observes is $9 * (L + H)$ time steps long. OOM refers to out of memory error. We observe that SNLDS provides a decent baseline, but is still underpowered compared to DeepTime, and other more approaches dedicated to long sequence time-series forecasting (e.g. Non-stationary Transformer baseline).
> >
> > UPDATE: We previously provided results for REDSDS [1] as a comparison to switching state space model, however, we found that the training is unstable, causing frequent errors, and thus provided results to SNLDS [2] instead.
> >
> > |  Methods |     |  DeepTime |           |   SNLDS   |           |  SNP  |       |
> > |:--------:|:---:|:---------:|:---------:|:---------:|:---------:|:-----:|:-----:|
> > |  Metrics |     |    MSE    |    MAE    |    MSE    |    MAE    |  MSE  |  MAE  |
> > |   ETTm2  |  96 | **0.166** | **0.257** |   0.231   |   0.308   | 3.156 | 1.377 |
> > |          | 192 | **0.225** | **0.302** |   0.264   |   0.315   | 3.189 | 1.399 |
> > |          | 336 | **0.277** | **0.336** |   0.320   |   0.351   | 3.264 | 1.414 |
> > |          | 720 | **0.383** | **0.409** |   0.435   |   0.420   | 3.203 | 1.381 |
> > |    ECL   |  96 | **0.137** | **0.238** |   0.293   |   0.359   | 0.984 | 0.814 |
> > |          | 192 | **0.152** | **0.252** |   0.283   |   0.356   | 0.976 | 0.813 |
> > |          | 336 | **0.166** | **0.268** |   0.324   |   0.386   | 0.982 | 0.814 |
> > |          | 720 | **0.201** | **0.302** |    OOM    |    OOM    | 0.969 | 0.811 |
> > | Exchange |  96 | **0.081** | **0.205** |   0.139   |   0.270   | 1.747 | 1.049 |
> > |          | 192 | **0.151** | **0.284** |   0.229   |   0.342   | 1.976 | 1.132 |
> > |          | 336 | **0.314** | **0.412** |   0.381   |   0.449   | 1.356 | 0.963 |
> > |          | 720 | **0.856** | **0.663** |   0.929   |   0.735   | 1.773 | 1.126 |
> > |  Traffic |  96 | **0.390** | **0.275** |   0.989   |   0.552   | 1.464 | 0.801 |
> > |          | 192 | **0.402** | **0.278** |   1.433   |   0.812   | 1.474 | 0.802 |
> > |          | 336 | **0.415** | **0.288** |    OOM    |    OOM    | 1.488 | 0.805 |
> > |          | 720 | **0.449** | **0.307** |    OOM    |    OOM    | 1.506 | 0.810 |
> > |  Weather |  96 | **0.166** | **0.221** |   0.217   |   0.271   | 0.633 | 0.601 |
> > |          | 192 | **0.207** | **0.261** |   0.265   |   0.306   | 0.601 | 0.593 |
> > |          | 336 | **0.251** | **0.298** |   0.314   |   0.336   | 0.638 | 0.606 |
> > |          | 720 | **0.301** | **0.338** |   0.382   |   0.377   | 0.654 | 0.626 |
> > |    ILI   |  24 |   2.425   |   1.086   | **2.240** | **0.937** | 6.249 | 1.720 |
> > |          |  36 |   2.231   |   1.008   | **2.219** | **0.942** | 6.031 | 1.680 |
> > |          |  48 | **2.230** | **1.016** |   4.209   |   1.498   | 5.133 | 1.560 |
> > |          |  60 | **2.143** | **0.985** |   4.141   |   1.474   | 5.279 | 1.563 |
> >
> > ---
> > [1] Ansari, A. F., Benidis, K., Kurle, R., Turkmen, A. C., Soh, H., Smola, A. J., ... & Januschowski, T. (2021). Deep Explicit Duration Switching Models for Time Series. Advances in Neural Information Processing Systems, 34, 29949-29961.
> >
> > [2] Dong, Z., Seybold, B., Murphy, K., & Bui, H. (2020, November). Collapsed amortized variational inference for switching nonlinear dynamical systems. In International Conference on Machine Learning (pp. 2638-2647). PMLR.

---

> ### Author Response · Authors · 2022-11-25
> **Further Discussions**
>
> Dear Reviewer i4vc,
>
> We hope that you have had the time to go through our response and revised manuscript. This is a gentle reminder and request, if our response has been satisfactory, that you consider raising your recommendation. If not, we look forward to further discussions with you.
>
> Best regards,
>
> Authors

---

### Official Review · Reviewer_NCFZ · 2022-10-28

**Confidence:** 2
**Correctness:** 3
**Technical Novelty And Significance:** 1
**Empirical Novelty And Significance:** 1
**Recommendation:** 3

**Clarity, Quality, Novelty And Reproducibility:**

Lack of clarity on motivation on why it is strong under non-strationtity and historical-based models are not sufficient. Poor mathematical description on problem formulation. Marginal novelty in chosing specific INR and hyperparameter for meta learning. only limited to multivarate modeling, not global modeling modeling even with its modeling capacity, where the baseline are known to be very poor.

**Strength And Weaknesses:**

## Strength
- enhanced deep-time class of method via INR and
- extensive multivariate real data experiments and ablation study


## Weakness
- clearly define what class of non-stationrity thei work target. It just generally mention covariate shift, conditional distribution shift, concept drift without formal definition. The figure 1 is not sufficient, except visuallizing locally-stationary (as high-level concept)
- need to provide more direct evidence on 'whereas existing neural forecasting methods, which are historical-value models, are unable to take full advantage of this formulation,' in the introduction
- need to clarify 'include having a stronger smoothness prior' in the abstract
- In figure 2, need to specify the method of naive deep-time method. At the same time, usually, historical-value based model can capture such a seonsonalty well.
- In section 2, what is the use of forecaster $h$?. the $\mathcal{L}$ is poorly defined, seems to $R^m \rightarrow R^m$ in eq (1) , which is different in problem formultation.
- What is $\Theta, \Phi$? how do the author decide? is it depending on model types? any rule of thumb? it is too general and vague with too much degree of freedom and little intuition.
- meta learning framework seesm to be nothing but a fitting some part of parameters like hyperaparemters in additional validation datasets. why this is specifically robust to non-stationary data?
- why only applied for long-term and multivariate forecaster? what about medium-term forecasting and univariate (global) forecasters  like DeepAR, MQ-RNN, ConvTrans, etc.
- lack of literature study on distribution shift, non-stationarity context. For example AdaRNN https://arxiv.org/abs/2108.04443

**Summary Of The Paper:**

This work developed light-weighted ts modeling, time-index meta learning scheme for time series forecasting.

**Summary Of The Review:**

Need to better highlight the motivation on why Deep-time class is considered over historical based classes, and why meta learning helps with theoretical analysis or strong intuition. The writing can be enhanced significally especially in problem formulation. More solid experiments beyond multivariate modeling can be done to demonstrate the need of this method.

---

> ### Author Response · Authors · 2022-11-18
> **Response to Reviewer NCFZ (3/3)**
>
> __Q10: Why is this only applied for long-term and multivariate forecasting?__
>
> A10: We first highlight that univariate forecasting results are included in Appendix J.
>
> Our paper aims to tackle the problem of non-stationarity for highly established Long Sequence Time-series Forecasting (LSTF) setting, which is a challenging benchmark for neural forecasting methods, built upon by several prior work [4,5,6,7,8] from reputable conferences (AAAI'21, NeurIPS'21, ICML'22, NeurIPS'22). Non-stationarity is a challenging problem in this setting due to the nature of the time-series datasets, for which the time-series are collected over a long period of time and experience concept drift.
>
> Global Forecasters are typically not considered, since they are specialized for global univariate datasets, meaning that each dimension of the time-series should represent the same measurement, but across different entities (i.e. measures temperature across different electricity transformers), datasets such as ETT do not fit this criteria, as each dimension of the dataset represents different measurements (i.e. temperature, load, etc. of the sample electricity transformer).
>
> __Q11: Lack of related literature on distribution shift and non-stationarity, for example, AdaRNN.__
>
> A11: Thank you for raising this point, we have added an extended related work section as Appendix A, which covers related literature on distribution shift and non-stationarity. We highlight AdaRNN, as well as a concurrent work, Non-stationary Transformer [8], which we have also added as a new baseline.
>
> ---
>
> [1] Bengio, Y., Courville, A., & Vincent, P. (2013). Representation learning: A review and new perspectives. IEEE transactions on pattern analysis and machine intelligence, 35(8), 1798-1828.
>
> [2] Finn, C., Abbeel, P., & Levine, S. (2017, July). Model-agnostic meta-learning for fast adaptation of deep networks. In International conference on machine learning (pp. 1126-1135). PMLR.
>
> [3] Dahlhaus, R. (2012). Locally stationary processes. In Handbook of statistics (Vol. 30, pp. 351-413). Elsevier.
>
> [4] Zhou, H., Zhang, S., Peng, J., Zhang, S., Li, J., Xiong, H., & Zhang, W. (2021, May). Informer: Beyond efficient transformer for long sequence time-series forecasting. In Proceedings of the AAAI Conference on Artificial Intelligence (Vol. 35, No. 12, pp. 11106-11115).
>
> [5] Challu, C., Olivares, K. G., Oreshkin, B. N., Garza, F., Mergenthaler, M., & Dubrawski, A. (2022). N-hits: Neural hierarchical interpolation for time series forecasting. arXiv preprint arXiv:2201.12886.
>
> [6] Wu, H., Xu, J., Wang, J., & Long, M. (2021). Autoformer: Decomposition transformers with auto-correlation for long-term series forecasting. Advances in Neural Information Processing Systems, 34, 22419-22430.
>
> [7] Zhou, T., Ma, Z., Wen, Q., Wang, X., Sun, L. &amp; Jin, R.. (2022). FEDformer: Frequency Enhanced Decomposed Transformer for Long-term Series Forecasting. <i>Proceedings of the 39th International Conference on Machine Learning</i>, in <i>Proceedings of Machine Learning Research</i> 162:27268-27286
>
> [8] Yong Liu, Haixu Wu, Jianmin Wang, and Mingsheng Long. Non-stationary transformers: Exploring the stationarity in time series forecasting. In Alice H. Oh, Agarwal, Danielle Belgrave, and Kyunghyun Cho (eds.), Advances in Neural Information Processing Systems, 2022. URL https://openreview.net/forum?id=ucNDIDRNjjv.

---

> ### Author Response · Authors · 2022-11-18
> **Response to Reviewer NCFZ (2/3)**
>
> __Q6: What is the use of forecaster $h$?__
>
> A6: Thank you for raising this important issue. In the original submitted manuscript, __Problem Formulation__ described the standard problem of forecasting, where $h: \mathbb{R}^{L\times m} \to \mathbb{R}^{H \times m}$ describes a historical-value model that takes as input a lookback window of time-series values, and predicts the forecast horizon. Our original intention was to describe the standard problem setting, and explain how to leverage a time-index model $f$, as a forecaster, $h$ (via our meta-learning formulation).
>
> We have updated the manuscript to directly focus on our problem setting, where we focus on the time-index model, $f$, and adaptation to the lookback window of time-index features and time-series values. We hope this makes the problem setting much clearer.
>
> __Q7: The loss $\mathcal{L}$ seems poorly defined, $\mathcal{L}$ in the problem formulation is different from $\mathcal{L}$ in equation (1).__
>
> A7: Thank you for raising this issue. As mentioned in the previous question/answer, the __Problem Formulation__ was previously focused on the standard time-series forecasting setting, where $\mathcal{L}$ referred to a forecasting loss for historical-value models. On the other hand, $\mathcal{L}$ in equations (1,2) referred to a reconstruction loss for time-index models. With our updated formulation, $\mathcal{L}$ now refers only to the reconstruction loss.
>
> __Q8: How are the parameterizations of $\Theta$ and $\Phi$ decided?__
>
> A8: Section 2.1 Forecasting as Meta-Learning first presents the generic framework of forecasting as meta-learning on deep time-index models. The parameterizations of $\Theta$ and $\Phi$ are generically described for this framework
>
> The specific parameterizations are described in Section 2.2 Model Architecture, where we present the specific form of $f$, which is a ReLU Multi-layered Perceptron with a random Fourier features layer. We also present the specific split of model parameters into $\theta$ and $\phi$.
> Specifically in the paragraph __Differentiable Closed-form Solvers__, we state that $\phi = \{W^{(0)}, b^{(0)}, \ldots, W^{(K-1)}, b^{(K-1)}, \lambda\}$ and $\theta = \{W^{(K)}\}$.
> This is motivated by the need to make the meta-learning formulation extremely efficient, and this parameterization enables a closed-form solution for meta-learning.
>
> Thus, we propose both a generic formulation which allows to transform any deep time-index model into a forecaster, and also propose a specific implementation of the forecaster and the split between meta and base parameters.
>
> __Q9: Meta-learning framework seems to be nothing but fitting some parts of parameters like hyperparameters in additional validation datasets, exactly why does this make it robust to non-stationary data?__
>
> A9: Please see our response to all reviewers for an overview of how DeepTime tackles non-stationarity. Below is specifics on how meta-learning tackles conditional distribution shift.
>
> In paragraph __Advantages of Deep Time-Index Meta-Learning__ of Section 1, we highlight several reasons as to why the meta-learning framework on deep time-index models help with non-stationarity. We expand on 2 of these critical reasons below:
>
> In general, meta-learning [2] is a framework to learn a model which can learn a general task distribution, which is then able quickly adapt to different tasks sampled from that task distribution. The parameters of the model which learns the general task distribution is known as the meta parameters, while the parameters which quickly adapt to different tasks is known as the base parameters.
>
> In the context of DeepTime, meta-learning solves the problem of conditional distribution shift, i.e. $p_t(y_{t+1}|y_t, y_{t-1}, \ldots) \neq p_{t'}(y_{t'+1}|y_{t'}, y_{t'-1}, \ldots)$, where $t \neq t'$. We follow the assumption of a locally stationary distribution [3], meaning that $p_t(y_{t+1}|y_t, y_{t-1}, \ldots) \approx p_{t'}(y_{t'+1}|y_{t'}, y_{t'-1}, \ldots)$ holds for $|t - t'| \leq \delta$ and $\delta$ is small. In words, this means that time steps which are nearby each other follow the same conditional distribution, whereas time steps which are far away do not have the same conditional distribution. Thus, in the language of meta-learning, each locally stationary distribution is a task, and the long time-series dataset contains a number of tasks.
>
> Meta-learning helps to overcome the problem of conditional distribution shift by learning meta parameters $\phi$, which can be seen as learning the task distribution. However, each task generated by this task distribution can be different, thus, this is where learning the base parameters, $\theta$, comes in, which can be seen as adapting to each task after observing some samples from the task (the lookback window).

---

> ### Author Response · Authors · 2022-11-18
> **Response to Reviewer NCFZ (1/3)**
>
> Thank you to Reviewer NCFZ for taking the time to read our work and give feedback and comments. We understand that the reviewer has raise concerns regarding the clarity on the writing.
> In our revised manuscript, we have revised (1) motivation of DeepTime over historical-based models, (2) problem formulation, for greater clarity based on the given feedback.
> We have also further elaborated about the definitions of non-stationarity and benefits of deep time-index meta-learning over historical-value models for non-stationary forecasting in the specific questions and response to all reviewers for your convenience.
> Finally, we highlight that our work builds upon the highly popular Long Sequence Time-series Forecasting setting [4,5,6,7,8] (ICML'22, NeurIPS'22, ...) which targets multivariate and univariate forecasting settings. Our empirical evaluations compares DeepTime against the current state-of-the-art in this field of research, on the exact same benchmark. We respectfully argue that this should fully demonstrates the need of our proposed method.
> We look forward to hearing any further feedback and are committed to address any remaining issues.
>
> __Q1: Clearly define what class of non-stationarity the work targets.__
>
> A1: Thank you for raising this issue up, please see our response to all reviewers which we address this issue.
>
> __Q2: Provide more direct evidence that "existing neural forecasting methods, which are historical-value models, are unable to take full advantage of this formulation".__
>
> A2: We clarify that our "forecasting as meta-learning" formulation is only defined on time-index models, and not on historical-value models. This is because we define the meta-learning formulation on input/output pairs of $(\tau_t, y_t)$ time-index and time-series value.
>
> While a similar formulation can be imagined for historical-value model input/output pairs, they will suffer from extreme sample efficiency. As mentioned in the paragraph titled __Advantages of Deep Time-index Meta-learning__ (Section 1), for a lookback window of length $L$ and forecast horizon of length $H$, a historical-value model requires N+L+H-1 time steps to construct a support set of size $N$ (compared to $N$ time steps for a time-index model). This problem is also much more computationally heavy, requiring $N$ forward passes through heavy models such as Transformer-based methods in order to make a single prediction.
>
> __Q3: What does a "stronger smoothness prior" refer to, as mentioned in the abstract?__
>
> A3: A "stronger smoothness prior" refers to one of the desirable properties for representation learning as described in Bengio's survey [1]. The property of smoothness assumes the function to be learned $f$ is such that similar inputs $x \approx y$ generally implies similar outputs $f(x) \approx f(y)$, which is described by Bengio as one of the most basic priors in machine learning.
>
> In our work, described towards the end of the paragraph __Advantages of Deep Time-index Meta-learning__ of Section 1, time-index models have a stronger smoothness prior because $t \approx t' \implies \tau_t \approx \tau_{t'} \implies f(\tau_t) \approx f(\tau_{t'})$.
>
> __Q4: In figure 2, what does the naive deep time-index model refer to?__
>
> A4: Figure 2 illustrates forecasts from (i) a deep time-index model trained on standard supervised learning formulation, and (ii) a deep time-index model endowed with our proposed meta-learning formulation, which we call DeepTime. We first reiterate, that a deep time-index model is merely a neural network (typically an MLP) which maps a time-index, to the value at that time-index, $f: \tau_t \mapsto y_t$. This deep time-index model can be trained ___naively___, via a standard supervised learning formulation denoted $\min_{\theta} \sum_{t=1}^{L} \mathcal{L}(f(\tau_t; \theta), y_t)$. This is contrasted to a deep time-index model trained via the meta-learning formulation as in Section 2.1, Equations (1, 2), a.k.a. DeepTime.
>
> __Q5: In figure 2, historical-value based models can already capture such patterns?__
>
> A5: Indeed, historical-value based models can already capture such patterns. The purpose of figure 2 is to highlight the differences between a naive deep time-index model, versus a deep time-index model endowed with the forecasting as meta-learning formulation (i.e. DeepTime).

---

> ### Author Response · Authors · 2022-11-25
> **Further Discussions**
>
> Dear Reviewer NCFZ,
>
> We hope that you have had the time to go through our response and revised manuscript. This is a gentle reminder and request, if our response has been satisfactory, that you consider raising your recommendation. If not, we look forward to further discussions with you.
>
> Best regards,
>
> Authors

---

> ### Author Response · Authors · 2022-12-05
> **Gentle reminder**
>
> Dear Reviewer NCFZ,
>
> Hope you are well! As the discussion period is coming to an end soon, we are looking forward to any feedback you have to our response despite your busy schedule. We will highly appreciate it if you let us know whether your previous concerns have been adequately addressed. Thank you once again.
>
> We have done our utmost to address the concerns as you suggested:
>
> 1. We have __included a formal definition and presentation__ on covariate and conditional distribution shift, and included an __extensive empirical analysis__ to verify the non-stationarity of real-world datasets
> 2. We have __updated the presentation__ of our proposed methodology, improving the notation based on the reviewer's valuable comments.
> 3. We have __clarified all queries__ regarding our method, comparisons, etc.
>
> Thank you again for your dedication to reviewing our paper and we are looking forward to your feedback.

---

> ### Author Response · Authors · 2022-12-07
> **Gentle reminder for further comments**
>
> Dear Reviewer NCFZ,
>
> Hope everything is going well. As the discussion period is coming to an end very soon, we would like to send a gentle reminder, and to let you know that we look forward to hearing any further updates and thoughts you may have. Once again, thank you for your efforts and service.

---

### Author Response · Authors · 2022-11-18
**Response to all reviewers (1/2)**

We thank all the reviewers for taking the time and effort to read through our work and giving comments and feedback.

The reviewers have raised some positive sentiments of our paper, expressing that our work explores ***an important and interesting direction of research*** (i4vc), and that our proposed framework ***is an interesting and canonical idea*** (comQ). Regarding our empirical evaluation, the reviewers thought that we have performed ***extensive multivariate real data experiments and ablation study*** (NCFZ), ***the results demonstrated improvement over a good number of baselines used*** (i4vc), and that ***the out-performance of other state-of-the-art transformers makes it a promising method*** (3pqc).

The reviewers also made constructive criticisms, which we have made all efforts to address, by providing additional empirical evidence, and by improving the clarity of our manuscript. We have highlighted the updates in our manuscript in blue, and summarize the changes we have made in the following:

1. ***Clarity of motivation, method, and definitions*** We have updated our manuscript to improve the clarity of our writing based on the reviewers' valuable feedback.
    * Improved motivation of proposing deep time-index models + meta-learning compared to historical-value models for non-statinoary forecasting, found in __Advantages of Deep Time-index Meta-learning__.
    * Improved clarity of problem formulation, focusing on defining the time-index meta-learning problem.
    * New section (Appendix B) covering formal definitions of covariate shift and conditional distribution shift.
2. ***Verification of non-stationarity of real-world datasets*** We have included in our manuscript, a new section (Appendix G.1) to verify the non-stationarity of the real-world datasets used for experiments, qualitatively, and quantitatively.
3. ***Additional baselines and ablations***
    * New relevant baseline (concurrent work), Non-stationary Transformer [1] which tackles non-stationarity for the Long Sequence Time-series Forecasting setting.
    * New "Extended Related Work" section (Appendix A), covering non-stationarity and distribution shift.
    * New sensitivity analysis on the effects of lookback length (Appendix L)
    * Additional ablations on variants of DeepTime trained by finetuning and full MAML (Appendix M)

Finally, we address some common questions that reviewers have raised below:

__Q1: Please provide a formal definition of "covariate shift" and "conditional distribution shift" which arise from the non-stationarity of time-series.__

A1: We have added the following as Appendix B of our updated manuscript:

In our work, we tackle the problem of non-stationarity in time-series data, which has been a well explored problem in the context of classical time-series analysis.
We map this problem to the modern setting of deep learning for time-series forecasting. As mentioned in Section 1 Introduction, long time-series datasets collected due to the advances of I.T. infrastructure has been plagued by the problem of non-stationarity. In particular, we tackle the problems of ___covariate shift___, and ___conditional distribution shift___ which arise from it.

__Definition 1 (Covariate Shift)__
Given a stochastic process $\{Y_t\}\_{t=1}^T$ and let $p(y_t, y\_{t-1}, \ldots, y\_{t-L+1})$ be the unconditional joint distribution of a length $L$ segment.
The stochastic process is said to experience covariate shift if any two segments are drawn from different distributions, i.e.
$p(y_t, y\_{t-1}, \ldots, y\_{t-L}) \neq p(y\_{t'}, y\_{t'-1}, \ldots, y\_{t'-L})$, $\forall \; t \neq t'$.

__Definition 2 (Conditional Distribution Shift)__
Given a stochastic process $\{Y_t\}\_{t=1}^T$ and let $p(y\_{t+1}|y_t, y\_{t-1}, \ldots, y\_{t-L+1})$ be the conditional distribution of $Y_{t+1}$ on a length $L$ segment of previous time steps.
The stochastic process is said to experience a conditional distribution shift if any two segments have different conditional distributions, i.e.
$p(y_{t+1}|y_t, y_{t-1}, \ldots, y_{t-L+1}) \neq p(y_{t'+1}|y_{t'}, y_{t'-1}, \ldots, y_{t'-L+1})$, $\forall \; t \neq t'$.

__Q2: How does DeepTime tackle the problems of covariate and conditional distribution shift?__

***Covariate shift*** has been shown to sharply degrade the prediction accuracy of deep learning models [2]. Historical-value models suffer from this failure mode, since they take as inputs time-series values, $y_t$. On the other hand, DeepTime, a time-index model, easily sidestep this problem by taking as input time-index features, $\tau_t$.

***Conditional distribution shift*** Following the assumption of a locally stationary distribution, where nearby time steps are assumed to follow the same data generating distribution [3] (see Figure 1), to be considered a task. The base learner adapts to this locally stationary distribution, while the meta learner generalizes across various task distributions.

---

> ### Author Response · Authors · 2022-11-18
> **Response to all reviewers (2/2)**
>
> __Q3: Please verify the non-stationarity of real-world datasets used in experiments.__
>
> A3: Real-world datasets used in long sequence time-series forecasting suffers from non-stationarity. In fact, this has also been studied by a concurrent work [1] on the same datasets.
>
> We first verify this __qualitatively__ by visualizing histogram values across some dimensions for each dataset in __Figure 6, Appendix G.1 of the updated manuscript__. This visualization already gives us a strong confirmation on the distribution mismatch between the training and testing phases.
>
> We further verify this __quantitatively__ via two statistical tests, the Augmented Dickey-Fuller (ADF) test, and the Chow test.
> * The ADF test helps verify ***covariate shift***. It tests the null hypothesis that a unit root is present in a time series sample. Not rejecting the null hypothesis indicates that a unit root is present, and is thus non-stationary.
> * The Chow test helps verify ***conditional distribution shift***. It is a test of whether the true coefficients in two linear regressions on different data sets are equal. Rejecting the null hypothesis of equality of regression coefficients in the two periods indicates that the train and test regions are generated from different distributions.
>
> Presented in the below table along with some dataset statistics, we report the results of both tests and the number of dimensions which meet the criteria for non-stationarity (rejecting the null hypothesis for Chow test, and not rejecting the null hypothesis for the ADF test) over various significance levels. We observe that the real-world datasets exhibit high levels of non-stationarity across dimensions based on both tests.
>
> |                          Dataset                  |    ETTm2   |     ECL    |   Exchange   | Traffic |    Weather   |   ILI  |
> |:-------------------------------------------------:|:----------:|:----------:|:------------:|:-------:|:------------:|:------:|
> |                     Dimensions                    |      7     |     321    |       8      |   862   |      21      |    7   |
> |                     Timesteps                     |   57,600   |   26,304   |     7,588    |  17,544 |    52,696    |   966  |
> |                 Sampling Frequency                | 15 minutes |   1 hour   |     1 day    |  1 hour |  10 minutes  | 1 week |
> |  Chow Test: # dims reject null   hypothesis @ 0.1 |      7     |     321    |       6      |   792   |      19      |    5   |
> | Chow Test: # dims reject null   hypothesis @ 0.05 |      7     |     320    |       5      |   771   |      19      |    5   |
> | Chow Test: # dims reject null   hypothesis @ 0.01 |      7     |        317 |       4      |   734   |      19      |    3   |
> |      Chow Test: average Chow test   statistic     |    49.13   |    23.55   |     21.25    |  16.93  |        25.39 |  2.77  |
> |      ADF Test: # dims fail to   reject @ 0.1      |      2     |     139    |       8      |   807   |       9      |    0   |
> |      ADF Test: # dims fail to   reject @ 0.05     |      2     |     120    |       8      |   739   |       7      |    0   |
> |      ADF Test: # dims fail to   reject @ 0.01     |      2     |     103    |       8      |   657   |       6      |    0   |
> |       ADF Test: average ADF test   statistic      |    -4.48   |    -5.24   |        -2.40 |  -3.29  |     -4.49    |  -6.89 |
>
> ---
>
> [1] Yong Liu, Haixu Wu, Jianmin Wang, and Mingsheng Long. Non-stationary transformers: Exploring the stationarity in time series forecasting. In Alice H. Oh, Agarwal, Danielle Belgrave, and Kyunghyun Cho (eds.), Advances in Neural Information Processing Systems, 2022. URL https://openreview.net/forum?id=ucNDIDRNjjv.
>
> [2] Nado, Z., Padhy, S., Sculley, D., D'Amour, A., Lakshminarayanan, B., & Snoek, J. (2020). Evaluating prediction-time batch normalization for robustness under covariate shift. arXiv preprint arXiv:2006.10963.
>
> [3] Dahlhaus, R. (2012). Locally stationary processes. In Handbook of statistics (Vol. 30, pp. 351-413). Elsevier.

---

### Decision · Program_Chairs · 2023-01-20

**Decision:**

Reject

**Justification For Why Not Higher Score:**

All reviewers indicate:
(i) a lack of clear definition of covariate and conditional distribution shifts in the paper. Time-series data has several drifts (covariate, prior probability, conditional probability, concept etc). Esp for the conditional distribution drifts, it is unclear if the paper is addressing the prior probability drift or the conditional probability drift.
(ii) a lack of clear definition of tasks,
(iii) a lack of empirical evidence to substantiate the claims made, while the paper is heavily motivated.

Thus, all reviewers have voted for a reject and this AC encourages the authors to take the reviewer's feedback and improve the work.

**Justification For Why Not Lower Score:**

N/A

**Metareview: Summary, Strengths And Weaknesses:**

The paper presents a deep time-index meta-learning  for non-stationary time-series data. It argues that meta-learning helps to alleviate co-variate shift, while it addresses conditional distribution shifts.

All reviewers indicate:
(i) a lack of clear definition of covariate and conditional distribution shifts in the paper. Time-series data has several drifts (covariate, prior probability, conditional probability, concept etc). Esp for the conditional distribution drifts, it is unclear if the paper is addressing the prior probability drift or the conditional probability drift.
(ii) a lack of clear definition of tasks,
(iii) a lack of empirical evidence to substantiate the claims made, while the paper is heavily motivated.

Thus, all reviewers have voted for a reject and this AC encourages the authors to take the reviewer's feedback and improve the work.





**Summary Of Ac-Reviewer Meeting:**

N/A